# Rift linkage and inheritance determine collisional mountain belt evolution

Sebastian G. Wolf [1] ✉, Ritske S. Huismans [1], Josep Anton Muñoz [2] & Dave A. May [3]

Many mountain belts - such as the Pyrenees, European Alps, Greater Caucasus, or Atlas - form through inversion of pre-collisional extensional basins. These orogens exhibit three-dimensional complexity, with along-strike variations in topography, orientation, and deformation patterns. Yet, the relationship between these characteristics and the inherited extensional architecture remains enigmatic. Here, we use 3D geodynamic models coupled to a landscape evolution model to investigate how pre-collisional rift-linkage controls rift inversion and mountain belt evolution. Our results show that initial mountain belt structure reflects the inherited basin geometry, while later growth is governed by subduction polarity. This polarity depends on the magnitude of basin offset and the presence of pre-existing structural weaknesses. Comparison with natural examples suggests that along-strike variations observed in the Greater Caucasus, Atlas, and Pyrenees can be explained by the inversion of segmented and offset rift basins. Building on these insights, we propose a simple diagnostic framework that establishes a direct link between topography and deep lithospheric structures, showing how extensional inheritance influences mountain building on Earth.

Mountain belts forming through continent-continent collision are one of the fundamental expressions of plate tectonics and are shaped by the interplay between tectonics, surface processes, and climate[1–4]. As part of the Wilson cycle[5], many collisional mountain belts form at the locations of previous rift systems that are inverted during orogenesis and now constitute a significant part of the orogen. Examples in a continent-continent collision setting are the Pyrenees[6,7], the Atlas mountains[8,9], the Greater Caucasus[10], the European Alps[11], and the Bowen basin in Australia[12]. Inversion of extensional basins is also prominent in ocean-continent collision systems, as for instance observed along-strike in the Andes[13–15]. Looking in plan view at collisional mountain belts on Earth, we observe that mountain belts are not linear features, but typically contain steps with variations in local orogen orientation (1), and branching into several mountain belt domains separated by lowlands (2) (Fig. 1a, b). Looking in plan-view at active continental rifts like the East African rift system, we observe that also rifts are not linear features (Fig. 1c). Rather, extensional systems

typically show offset basins and grabens that are often linked and connected through transfer zones, creating discrete steps (i), and branching of one rift arm into several domains (ii). A reconstruction of the pre-extensional template in the Pyrenees also exhibits linked and offset rift basins with discrete steps (i) and several rift branches (ii), which were reactivated and inverted during orogenesis (Fig. 1d). The observation of an ubiquitous extensional history pre-dating mountain building, and the observations that extensional systems and mountain belts show similar 3D complexities poses the question: How does the 3D architecture of pre-collisional extension influence mountain building?

Geodynamic modelling studies investigating the dynamics of inversion-tectonics during orogenesis found that extensional inheritance can have a strong influence on the structural evolution of mountain belts[16–23]. Most studies investigated the influence of extensional inheritance in a 2D cross section and found that weak inherited extensional structures are preferentially reactivated during collision

[1]Department of Earth Science, University of Bergen, Bergen, Norway. [2]Geomodels Research Institute, University of Barcelona, Barcelona, Spain. [3]Institute of Geophysics and Planetary Physics, Scripps Institution of Oceanography, University of California San Diego, La Jolla, CA, USA. ✉e-mail: sebastian.wolf@uib.no

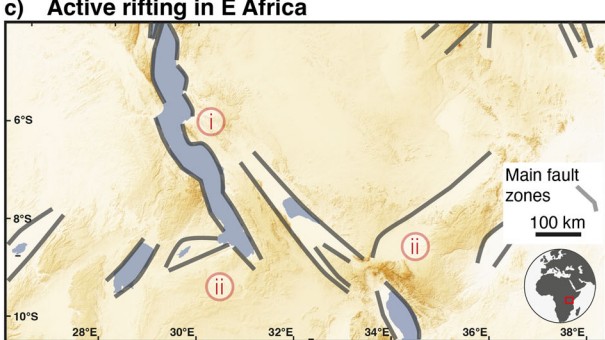

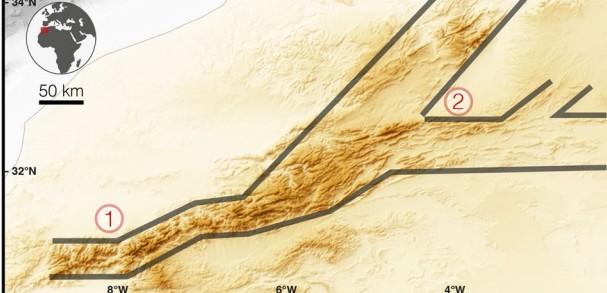

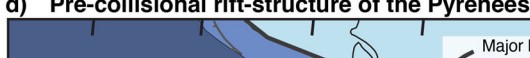

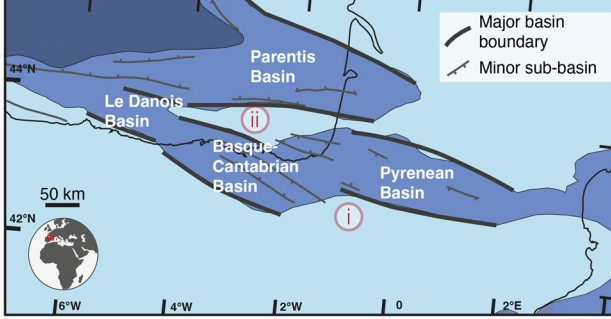

**Fig. 1 | Plan-view structure of two mountain belts and two extensional systems on Earth. a** Topographic map of the Greater Caucasus with first-order outline higlighting the stepping (1) nature of the mountain belt. **b** Topographic map of the Atlas with outline highlighting the stepping (1) and branching (2) mountain belt structure. **c** Topographic map of the western part of the East African rift system with an outline of the main structural grain based on active faulting[69] highlighting the stepping (i) and branching (ii) of this continental rift. **d** Reconstructed Cretaceous pre-collisional rift structure in the Pyrenees following[45], showing an off-set, stepping (i) and branching (ii) extensional system. DEMs in (**a**–**c**) are from ETOPO22[70] and have the same colourmap as shown in (**a**).

and determine the structural style in a mountain belt[16–21,23]. 3D studies of inversion tectonics primarily focused on the brittle part of the lithosphere and showed how inversion of extensional basins influences the evolution of fold-thrust belts, see review in ref. 22. These predominantly analogue modelling studies often exclude mantle dynamics and are inherently constrained by material properties, limiting their ability to reproduce inherited extensional structures. Hence, the importance of a 3D rift setting with stepping, offset, and linked rift basins on inversion tectonics remains unknown when including a consistent upper mantle-scale perspective.

Here, we use 3D upper-mantle-scale geodynamic models coupled to a surface processes model to investigate how rift linkage influences subsequent inversion during continental collision and mountain building. We choose a relatively simple model setup with two offset extensional basins that are subsequently inverted during continent-continent collision. The aim of this article is (a) to provide key characteristics of 3D inversion tectonics during mountain building, that can be used as a simple template to explain mountain belts on Earth, (b) to quantify the underlying physical principles that guide model evolution, and (c) to apply model results to well-studied mountain belts, specifically the Greater Caucasus, Atlas, and Pyrenees.

## Results
### Modelling results
We developed a coupling between the 3D thermo-mechanical geodynamic model pTatin3D[24,25] and the surface processes model FastScape[26,27], to investigate the effects of extensional inheritance on mountain building and the associated surface processes response. The tectonic model is laterally uniform and filled with materials corresponding to a typical layered continental lithosphere on Earth (see "Methods" section). The surface processes model solves for fluvial erosion, hillslope creep and sediment deposition. We do not vary the efficiency of fluvial erosion, but choose constant parameters for all

models. The mountain belts produced by the chosen set of parameters can be classified as tectonics-dominated mountain belts with Beaumont number $Bm = 3.5$, i.e., mountain belts that do not reach flux steady state, which is the dominant mountain belt type on Earth[4]. See Methods section for computation of $Bm$.

Model deformation is localised and initiated using two weak seeds. To investigate the influence of pre-collisional offset rift basins on mountain building, we vary the offset between the weak seeds and the shape of the seeds. Each model is extended for 12 Myr with a rate of 1 cm/yr in the east-west direction, followed by a reversal of the velocity boundary conditions and shortening with the same rate. This period of extension leads to full crustal break-up; the applied rate of extension/shortening represents typical plate velocities on Earth. Models run for 63 Myr but were stopped when the evolving orogen reached the eastern or western model side. This period and rate of shortening create orogens reaching intermediate size, slightly larger than, for instance, the European Alps. We first present the reference model without any offset between the seeds (M1), followed by a model with a very large offset of 400 km (M2), and a model with an intermediate offset of 200 km (M3). Finally, we present one model with an intermediate offset between the weak seeds (200 km) but with a preferential orientation of elongated weak seeds that mimic a pre-existing weak fabric (M4). Supplementary model output showing deformational regimes and deviatoric stresses is presented in Supplementary Figs. S1–S4.

Model 1 (Fig. 2) has simple and uniform weak seeds without offset between them. During extension, deformation first localises at the weak seeds and forms two grabens that quickly propagate and link to form one cylindrical rift. At the end of the extension phase, the continuous rift system shows full crustal break-up with more offset in the weak seed domains than in the model centre (Fig. 2a). The symmetric rift faults are reactivated during the inversion stage and form a small juvenile mountain belt with pieces of (sub-)lithospheric

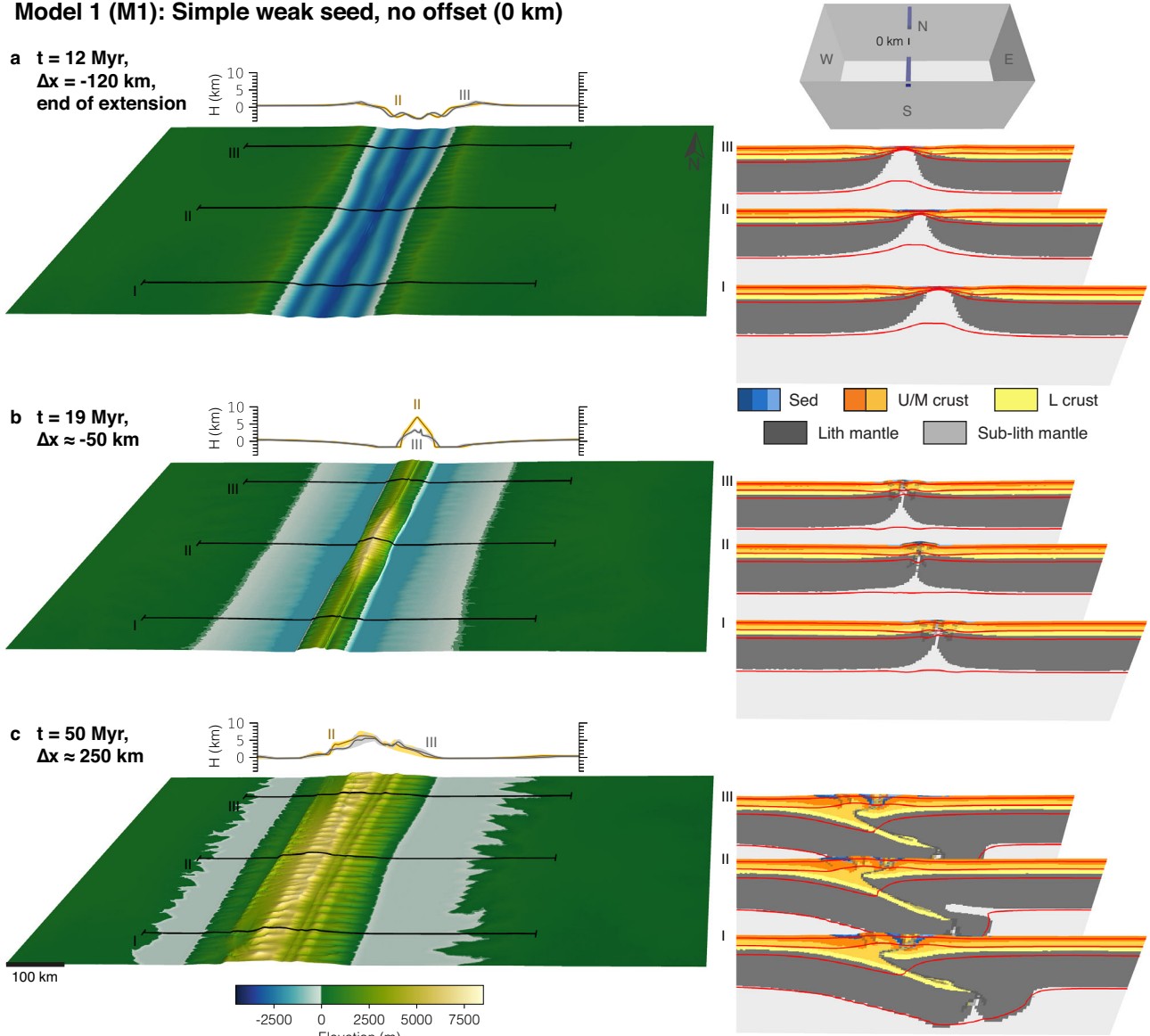

**Fig. 2 | Model 1 with 0 km offset and simple seeds. a** Snapshot at the end of the rifting phase. **b** Snapshot during the initial inversion stage. **c** Snapshot during the mature mountain building phase. The small inset in the upper right corner shows the initial weak-seed configuration of the model with simple uniform seeds with 0 km offset. Each sub-figure consists of the model surface to the left with three corresponding cross-sections shown to the right. Behind the model surface, two elevation-swath profiles at II and III are shown. Abbreviations for model materials are as follows: Sed are sediments deposited during runtime; U/M crust or L crust are Upper/Middle and Lower crust, respectively; (Sub-)Lith mantle are the lithospheric and sub-lithospheric mantle. Red lines are temperature contours at 100 °C, 350 °C, 550 °C, and 1330 °C. Strain-weakened fault zones in the model cross sections are indicated by a grey semi-transparent overlay.

mantle trapped in the inverted domain (Fig. 2b, cross sections). Shortening and topography are highest in the central domain, which experienced the least amount of extension. During the ensuing mountain building stage (Fig. 2c), the model grows by subduction of the western lithospheric mantle, and formation of thick-skinned crustal thrust sheets originating from the western, pro-side of the orogen. This mature inversion orogen is translated towards the pro-side of the orogen and keeps the cylindrical structure inherited from the rifting phase. The landscape is dominated by rivers flowing perpendicular to the orogen strike. Throughout the manuscript, we refer to 'juvenile' and 'mature' rifts or orogens. Juvenile rifts contain unconnected sub-basins, unlike mature rifts, where these are linked (Fig. 2a). Juvenile orogens lack clear subduction polarity and are only composed of inverted extensional structures (Fig. 2b), whereas mature orogens exhibit distinct lithospheric subduction and additional crustal shortening (Fig. 2c).

Model 2 (Fig. 3) is characterised by the same initial and boundary conditions as Model 1, but with a large offset between the weak seeds of 400 km. Initially, extension creates two isolated rift basins at the weak seeds. After around 6 Myr, the eastern rift basin becomes aborted, and the western basin propagates to the southern boundary of the model and turns into the main locus of extension (see Supplementary Movie 2). At the end of the extension phase, the western rift shows full crustal break-up near the northern boundary, and less extension near the southern boundary (Fig. 3a). During the inversion phase, the western rift is the main locus of deformation, but the eastern, aborted rift also gets inverted and forms a small and low mountain belt. Again, small pieces of lower crust and lithospheric mantle are trapped in the inverted domain (Fig. 3b). With ongoing convergence, the western, main mountain belt behaves similarly to Model 1: It grows by one-sided subduction and formation of thick-skinned thrust sheets originating primarily from the pro-side with translation of the orogen towards the

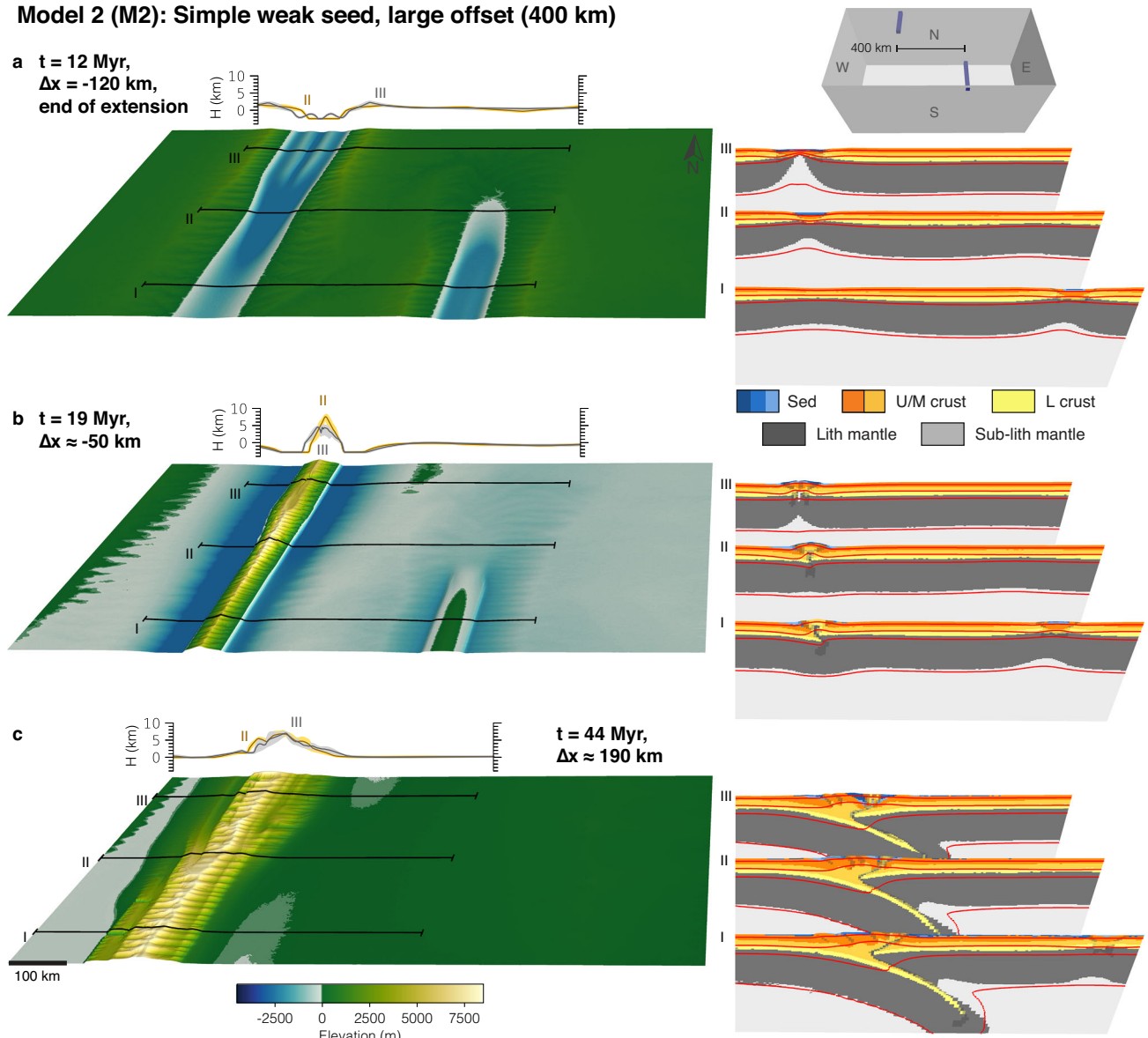

**Fig. 3 | Model 2 with 400 km offset and simple seeds. a** Snapshot at the end of the rifting phase. **b** Snapshot during the initial inversion stage. **c** Snapshot during the mature mountain building phase. The small inset in the upper right corner shows the initial weak-seed configuration of the model with simple uniform seeds with 400 km offset. Each sub-figure consists of the model surface to the left with three corresponding cross-sections shown to the right. Behind the model surface, two elevation-swath profiles at II and III are shown. Abbreviations for model materials are as follows: Sed are sediments deposited during runtime; U/M crust or L crust are Upper/Middle and Lower crust, respectively; (Sub-)Lith mantle are the lithospheric and sub-lithospheric mantle. Red lines are temperature contours at 100 °C, 350 °C, 550 °C, and 1330 °C. Strain-weakened fault zones in the model cross sections are indicated by a grey semi-transparent overlay.

west. The growing orogen is cylindrical with river flow perpendicular to the orogen boundary, and mimics the pre-existing extensional structure (Fig. 3c).

Model 3 (Fig. 4) is characterised by the same boundary and initial conditions as Model 1, with the difference of an intermediate offset of the weak seeds of 200 km. Extension first localises at the two weak seeds, forming two offset rift basins. With ongoing extension, the two rift basins propagate and link after 6 Myr of model evolution. Subsequently, they form an offset, linked and continuous rift basin with enhanced extension in the initial rift basins compared to the model centre (Fig. 4a). The 'inner faults' of the offset rift basin, i.e., the eastern fault in the northern basin and the western fault in the southern basin record more offset and therefore also slightly higher rift-flank topography (Fig. 4a). During the inversion stage, the extensional structures are reactivated and form a small and continuous mountain belt that

has the same plan-view shape as the linked rift basin. The primary orientation of the drainage networks is perpendicular to the local strike of the orogen, so that the first-order river orientation is changing along-strike (Fig. 4b). Continued shortening of the orogen leads to the development of opposite-polarity subduction, and linkage of the two inner faults of the former rift basin. During growth, the orogen increases in size by thick-skinned thrusting, with thrust sheets forming on the respective pro-sides of the orogen. This mechanism leads to orogen growth and translation towards the west in the southern part of the model, and growth and translation towards the east in the northern part, reversing the inherited rift structure. Interaction of the two subducting slabs is characterised by several large pieces of lower crust and lithospheric mantle sheared off from the opposite-polarity subducting lithospheres, and creating a topographic low in the transfer zone (Fig. 4c).

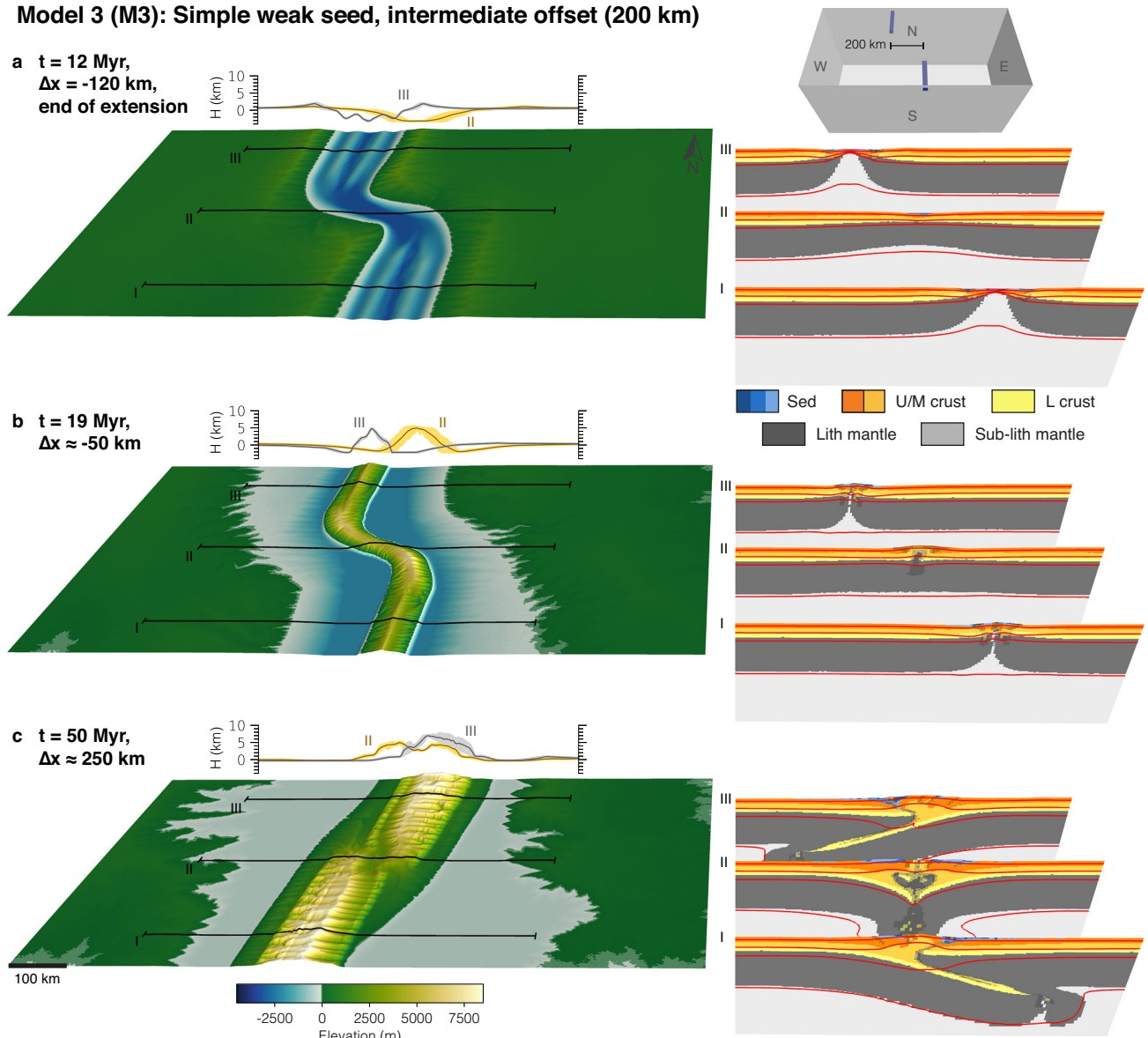

**Fig. 4 | Model 3 with 200 km offset and simple seeds. a** Snapshot at the end of the rifting phase. **b** Snapshot during the initial inversion stage. **c** Snapshot during the mature mountain building phase. The small inset in the upper right corner shows the initial weak-seed configuration of the model with simple uniform seeds with 200 km offset. Each sub-figure consists of the model surface to the left with three corresponding cross-sections shown to the right. Behind the model surface, two elevation-swath profiles at II and III are shown. Abbreviations for model materials are as follows: Sed are sediments deposited during runtime; U/M crust or L crust are Upper/Middle and Lower crust, respectively; (Sub-)Lith mantle are the lithospheric and sub-lithospheric mantle. Red lines are temperature contours at 100 °C, 350 °C, 550 °C, and 1330 °C. Strain-weakened fault zones in the model cross sections are indicated by a grey semi-transparent overlay.

Model 4 (Fig. 5) is characterised by the same boundary and initial conditions, and intermediate seed offset as Model 3, but contains two elongated and angled weak seeds that dip in the same direction. This model setup mimics a pre-existing weak fabric with homogeneous along-strike orientation. During extension the model evolves similarly to Model 3, with the difference of larger margin asymmetry related to the shape of the weak seeds (Fig. 5a). Model evolution is also similar to M3 during the inversion stage, forming a continuous mountain belt with the same plan-view shape as the pre-existing extensional basins and river-flow perpendicular to the local strike of the orogen (Fig. 5b). With ongoing convergence, the model develops same-polarity subduction. Same-polarity subduction is facilitated by the linkage of the eastern basin bounding faults of the northern and southern sub-basins that are related to the pre-existing weak zones. The orogen grows by thick-skinned thrusting of thrust sheets forming on the pro-side of the

orogen, inducing orogen-translation towards the pro-side. To first order, the plan-view rift structure is maintained during orogenic growth, and the orogen continuously exhibits river flow perpendicular to the local orogen boundary orientation. As a second-order structural feature, we observe that a piece of lower crust and lithospheric mantle is sheared off in the southern part of the linkage zone (Fig. 5c, cross section II) and becomes trapped at shallow depths.

Supplementary Models 1–3 (SM1-3) show sensitivities to a very small seed offset (SM1), to an intermediate seed offset lower than in M3 (SM2), and to opposite polarity in dip of the weak seeds (SM3). SM1, with a very small offset between simple weak seeds of 20 km, shows similar behaviour to Model 1, with same-polarity subduction leading to first-order to a conservation of the offset basin structure during inversion and mountain growth (Supplementary Fig. S5). SM2, with an intermediate offset of 100 km between simple weak seeds, exhibits

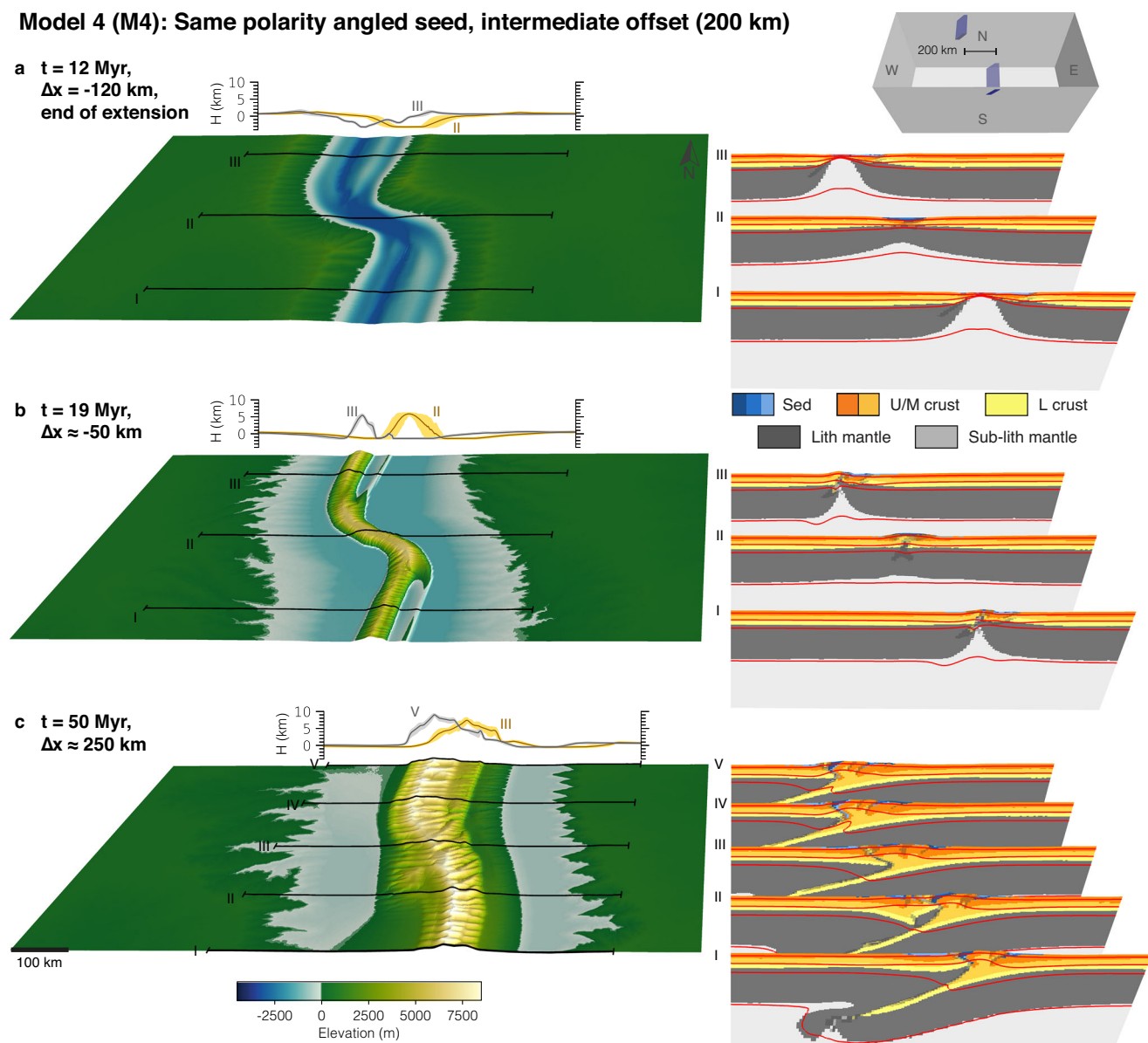

**Fig. 5 | Model 4 with 200 km offset and same-polarity angled seeds. a** Snapshot at the end of the rifting phase. **b** Snapshot during the initial inversion stage. **c** Snapshot during the mature mountain building phase. The small inset in the upper right corner shows the initial weak-seed configuration of the model with same-polarity angled seeds with 200 km offset. Each sub-figure consists of the model surface to the left with three or five corresponding cross-sections shown to the right. Behind the model surface, two elevation-swath profiles are shown.

Abbreviations for model materials are as follows: Sed are sediments deposited during runtime; U/M crust or L crust are Upper/Middle and Lower crust, respectively; (Sub-)Lith mantle are the lithospheric and sub-lithospheric mantle. Red lines are temperature contours at 100 °C, 350 °C, 550 °C, and 1330 °C. Strain-weakened fault zones in the model cross sections are indicated by a grey semi-transparent overlay.

very similar behaviour to M3, with opposite polarity subduction as a result of inner fault-linkage (Supplementary Fig. S6). Additional models with similar offsets all develop into opposite polarity subduction. These results show that opposite polarity subduction in intermediate-offset simple seed models is a systematic and repeatable model result. SM3, with the same initial and boundary conditions as Model 4 but with an opposite polarity in dip of the weak seeds, shows a very similar evolution as Model 3: Opposite polarity subduction develops through linkage of the inner faults that are also related to the inherited weaknesses (Supplementary Fig. S7).

## Model analysis
Our models show that rift basins tend to link when the lateral offset between them is small to moderate, but fail to link at large offsets,

which is consistent with previous work[28–31]. To explain this behaviour, we present a simple force-minimisation analysis that quantifies how rift linkage depends on offset magnitude. See supplementary information for detailed information on the definition of the different forces involved, including characteristic values and simplifying assumptions. We assume that deformation follows the path of least resistance, minimising total mechanical work, and that rifting starts in the southern weak zone (segment $P_1$-$P_2$, Fig. 6a). From this initial segment, deformation can follow two possible paths: (a) it links to a second weak zone and utilises the segmented path $D' + D_{seed}$, requiring a force $F_{Link}$, or (b) it bypasses the linkage and progresses directly to the model boundary along path $D_b$, requiring a force $F_{NoLink}$. Linkage occurs only when $F_{Link} < F_{NoLink}$. Since the direct path of (b) is shorter, linkage and hence path (a) will only be utilised if the energetic gain $G$ from

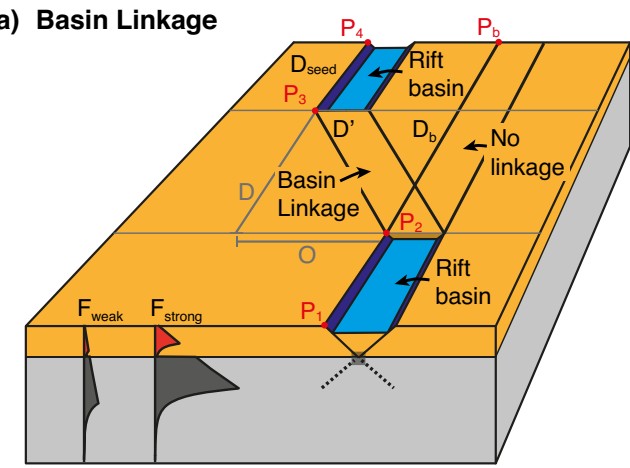

**a) Basin Linkage**

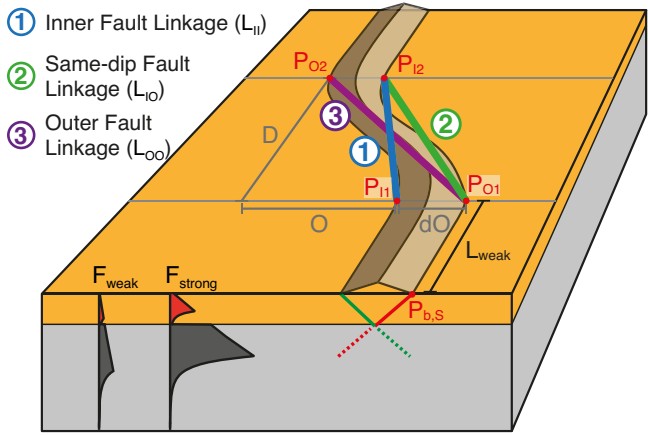

**b) Subduction polarity selection**

① Inner Fault Linkage ($L_{II}$)

② Same-dip Fault Linkage ($L_{IO}$)

③ Outer Fault Linkage ($L_{OO}$)

**Fig. 6 | Block diagrams explaining and quantifying model behaviour. a** Block diagram showing the possible paths of linkage during rifting of offset weak zones. **b** Block diagram showing the possible linkage options determining subduction-polarity during inversion of an extensional rift basin.

deforming along the weak zone exceeds the penalty $P$ of following a longer path. Applying these force considerations to Model 2, we can calculate $F_{Link} \approx 6.4 \times 10^{18}$N $> F_{NoLink} \approx 5.2 \times 10^{18}$N, which explains why the basins do not link. On the contrary, in Model 3 $F_{Link} \approx 4.4 \times 10^{18}$N $< F_{NoLink} \approx 5.2 \times 10^{18}$N, which explains the observed basin linkage. For our set of parameters the maximum offset for rift linkage to occur, i.e., $F_{Link} = F_{NoLink}$, is reached at 292 km offset.

To summarise, model behaviour can be successfully quantified by assuming that deformation follows the path of least resistance, such that the integral of the depth-integrated lithospheric strength along any given path is minimised. This implies that, in nature, weak zones in closest proximity are more likely to link during rifting. While this outcome may appear intuitive, the force-minimisation analysis offers a physically grounded explanation and predictive framework for understanding rift-basin linkage.

Using force minimisation principles, we can also quantify the evolution of subduction polarity in our models and its implications for orogenic systems on Earth. Consider the inversion of an intermediate-offset rift basin, as shown in Models 3 and 4 and illustrated in Fig. 6b. To first order, rifting produces a conjugate set of shear zones that dip toward the basin centre. During subsequent inversion, these shear zones are reactivated. As the lithosphere is restored to its normal thickness, the resulting mountain belt retains this conjugate shear zone geometry (represented in green and red in Fig. 6b). At this stage, deformation may proceed via three potential fault linkage pathways:

1. **Inner fault linkage** – connecting points $P_{I1}$ and $P_{I2}$ via segment $L_{II}$.
2. **Same-dip fault linkage** – connecting $P_{O1}$ and $P_{I2}$ via $L_{IO}$ (or equivalently $P_{I1}$ and $P_{O2}$)
3. **Outer fault linkage** – connecting $P_{O1}$ and $P_{O2}$ via $L_{OO}$

Minimising deformational work, it is expected that deformation utilises the shortest path, so that the inner faults link and $L_{II}$ is deformed (option 1). This linkage creates opposite polarity subduction related to the different dip-angles of the linked faults and can also be observed in Model 3. Linking faults with the same dip direction via $L_{IO}$, option 2, requires greater deformational work because of the longer path, incurring a mechanical penalty $P$. For this pathway to be favoured, it must yield a mechanical advantage, or gain $G$, such that $G > P$. One potential source of gain is a difference in fault strength. For example, if the west-dipping shear zone (of length $L_{weak}$ in Fig. 6b) is mechanically weaker than its east-dipping conjugate, it may be preferentially reactivated during shortening. This behaviour is observed in Model 4, where inversion proceeds via reactivation of the west-dipping shear zones. Here, the presence of pre-existing mechanical weaknesses provides sufficient gain to offset the additional work required by the longer deformation path. Finally, linkage of the two outer faults via $L_{OO}$, option 3, also leads to opposite polarity subduction, but involves the longest deformation path and highest mechanical cost, making it the least energetically favourable option.

These force minimisation principles also explain same-polarity subduction in cases of a straight rift system without significant offset, as in Models 1 and 2. In such configurations, $L_{II} > L_{IO}$, so the system preferentially reactivates one set of same-dipping faults (option 2), establishing a subduction interface with uniform polarity. Whether the left- or right-dipping faults are favoured depends on small perturbations during model evolution, such as localised erosion or variations in fault offset. The transition of the energetically most favourable linkage from same- to opposite-polarity subduction is governed by the magnitude of basin offset $O$. Once $O > 0.5 dO$, where $dO$ is the surface distance between the conjugate and inverted rift-bounding shear zones (approximately 60 km in our models), opposite polarity subduction becomes the favourable option. Accordingly, any linked basin with an offset greater than 30 km will favour opposite polarity subduction. This transition explains why supplementary Model 2 (20 km offset and simple weak seeds) exhibits same-polarity subduction, whereas Model 3 (200 km offset and simple weak seeds) evolves into opposite-polarity subduction.

To summarise, the default behaviour for inversion of linked basins with small offset is same-polarity subduction. Once basin offset is larger than a few tens of km, i.e., $L_{IO} > L_{II}$, the default model behaviour is opposite polarity subduction and linkage of the inner faults. If, however, there are inherited weaknesses with a preferential orientation that provide a gain over the penalty of longer linkage $G > P$, same-polarity subduction may still prevail.

**Comparison to the Greater Caucasus, Atlas, and Pyrenees**

We next compare our modelling results with three mountain belts on Earth that are well known for their pre-collisional extensional history: the Greater Caucasus, the Atlas, and the Pyrenees.

The Greater Caucasus is an actively growing orogen forming between the Black and Caspian Seas as a result of the Arabia-Eurasia collision (Fig. 7a). The orogen is primarily a result of inversion of the Greater Caucasus basin, a continental extensional back-arc basin that resulted from subduction of the Neotethys below Eurasia and separated the Transcaucasian-Lesser Caucasus (South Caucasus) from the Scythian Platform (North Caucasus)[32]. Back-arc rifting occurred during the Jurassic, which formed a hyper-extended continental rift system, and was followed by a period of tectonic quiescence and basin filling[10,33,34]. The onset of inversion is debated but happened during late Eocene-Oligocene times (~ 35 Ma–30 Ma)[33–35] and is likely related to the

## Natural examples of mountain belts forming by inversion of extensional structures

### a  Greater Caucasus

### b  Atlas

### c  Pyrenees

Syn-/Post-collision sedimentary rocks

Pre-collision sedimentary rocks

Upper/Lower Crust

Lithospheric Mantle

**Fig. 7 | Digital elevation models and cross sections of the Greater Caucasus, Atlas, and Pyrenees.** The DEMs are from ETOPO22[70]. In each DEM, the black line follows the drainage divide; the dashed lines are an offset of the black line. The dotted line in the Atlas delineates to first order the mountain belt orientation. **a** Cross-section through the Greater Caucasus is based on ref. 10, with deformation at depth > 25 km stipulated based on typical modelling results and inferred from refs. 37,38. DEM-areas marked with `B' are potential pre-collisional rift basins

connected by transfer zones 'T'. **b** The cross-section through the Central High Atlas is based on ref. 8. WHA is short for Western High Atlas, CHA is Central High Atlas, EHA is Eastern High Atlas, MA is Middle Atlas. **c** The left cross-section through the Basque Pyrenees is based on ref. 45; the middle section, ECORS-Arzacq, is based on ref. 71; the right section, ECORS-Pyrenees, is based on ref. 6. Labels on DEM show (1) the Labourd gravity anomaly, (2) the Bilbao gravity anomaly, (3) the transfer zone between the reconstructed Pyrenean and Basque-Cantabrian pre-collisional basins.

closure of the Neotethys and initiation of the Arabia-Eurasia collision[36]. The orogen is underlain by a N-dipping slab that is possibly tearing off in the W[37,38]. The slab length of 100–200 km indicates a similar amount of shortening of the mountain belt. Thickening and shortening are primarily located at the southern, pro-side, of the orogen[35,39]. The

orogen also exhibits deformation on the retro-side and several thin-skinned foreland fold-thrust belts detaching in shallow weak horizons[33,35,39]. Following the drainage divide of the mountain belt, we see that the orogen has an overall NW-SE trend and is compartmentalised into longer NW-SE trending domains (B) that are offset by

30–40 km and connected by shorter W-E-trending and NNW-SSE-trending segments (T; Fig. 7a). This segmentation can also be observed in the Main Central Thrust (MCT), which is inferred to constitute the boundary between the Pro- and Retro-sides[10]. The orogen has a continuous and high topography with maximum elevations > 5 km. The fluvial network follows, to first order, the changing trends in strike of the mountain belt, with drainage basins that are pre-dominantly perpendicular to the local orientation of the drainage divide. The Greater Caucasus with its continuous high topography, high amounts of shortening, drainage network orientation, extensional history and changes in orogen-strike fits very well with a mature inversion orogen that exhibits the same subduction-polarity along strike, as modelled here in Model 4 (Fig. 7a). This characterisation implies that the orogen-segmentation reflects the configuration of the pre-collisional extensional basins, which we interpret as several extensional sub-basins (B) that are connected by transfer zones (T; Fig. 7a). Our modelling and force balance analysis indicates that same-polarity subduction orogens are favoured by weak-zones with a preferential orientation that provide deformational gain (Model 4, Figs. 5, 6b). A reconstruction of the extensional structure describes several N-dipping detachment levels[10], which are likely reactivated during orogenesis and provide the weaknesses necessary to develop the observed same-polarity subduction orogen.

A well-known example of widespread basin inversion during continental orogenesis is the Atlas. The Atlas extends over 2000 km from the Moroccan coast through Algeria to Tunisia and the coastal Tell-mountains, and also branches out into the Rif-mountains through the Middle Atlas (Fig. 7b). Triassic and Jurassic rifting related to the opening of the Atlantic formed a linked intra-continental rift system that reactivated pre-existing Hercynian structures[40]. Shortening and inversion of the filled continental rift basins started in the Eocene with the main phase in the Oligocene and Miocene[9,41]. Shortening reactivated extensional structures, forming a mountain belt that is located on top of the pre-collisional basins[9,41,42]. Inversion was very mild with the highest amount of shortening of at most 35 km in the Central High Atlas, which potentially exhibits incipient subduction of lower crust and lithospheric mantle towards the north (Fig. 7b)[8,9,42]. Low amounts of shortening and exhumation explain why upper crustal rocks are in some locations at lower crustal levels inside the mountain belt compared to the forelands (Fig. 7b). Pre-collisional rifting was less pronounced in the Western High Atlas compared to the Central High Atlas, which explains exhumation of pre-collisional Palaeozoic rocks in the west[43]. Low topography in the eastern part of the Atlas indicates very low amounts of shortening and inversion in this domain. A relatively thin crustal root combined with a thinned mantle lithosphere suggests that some topography is supported by mantle processes[9,44]. Looking at the large-scale trends of the Atlas, we find that the western branch, the so-called High Atlas, trends SW-NE, with a change in strike at the Eastern High Atlas (EHA) towards W-E, which changes again after ~300 km to a SW-NE strike. The northern branch termed Middle Atlas (MA) has yet another orientation with a more northerly direction (Fig. 7b). The different large-scale domains exhibit heterogeneous orogen heights, with the High Atlas being a domain of continuous high topography. In the High Atlas, we observe a stepping drainage divide with river flow perpendicular to the local orientation of the divide, i.e., perpendicular to the local strike of the orogen. The mild inversion of extensional basins with low amounts of shortening and incipient subduction indicate that the Atlas is a juvenile inversion orogen where our models show that subduction polarity does not yet play a significant role, as shown in Models 3 and 4 after ~ 20 Myr of model evolution (Figs. 4b, 5b). The large-scale stepping of the mountain belt indicates that orogenesis reactivated a linked system of extensional basins. Different amounts of shortening and inversion, and potentially different amounts of mantle contribution, likely explain the variations in topography along the entire strike of the Atlas. Focussing only on

the High Atlas, we interpret the stepping of the mountain belt strike with associated changes in fluvial drainage orientation as resembling the previous small-scale rift structure, with local SW-NE-trending basins with more E-W-trending linkage zones. The Atlas is a fitting example of a juvenile inversion orogen where subduction polarity has no primary influence, and highlights that basin linkage and associated changes in mountain belt structure during inversion occur on different spatial scales, from small-scale steps (10s of km) to large-scale branching (100s of km).

A classical example of basin-inversion during orogenesis are the Pyrenees. This double-vergent orogen formed as a result of the Iberia-Eurasia collision between Late-Cretaceous (~ 85 Ma) and Miocene times (~ 20 Ma)[6,45,46]. Collision inverted a laterally non-uniform extensional system which resulted in different characteristics along strike (Fig. 1d). In the E, between France and Spain, the Pyrenees sensu stricto (s.s.) formed by continent-continent collision that inverted the Pyrenean extensional system and also propagated into the un-thinned Iberian pro-side (Figs. 1d, 7c)[6,7,47]. Collision was accommodated by N-ward subduction of the Iberian lower crust and lithospheric mantle and crustal thickening, with shortening values of at most 90–165 km, decreasing from E to W[6,7,46]. The structural style is strongly influenced by pre- and syn-collisional weak salt layers that facilitated a pro-wedge thin-skinned foreland fold thrust-belt, and induced antiformal stacking of thick-skinned thrust sheets in the orogen core (Fig. 7c, section C-C')[6,7,18,48]. Exhumed fragments of mantle rocks on the N-flank (retro-wedge) of the orogen, are found embedded in metamorphosed Mesozoic sedimentary cover between inverted crustal extensional blocks[49,50]. A large positive gravity anomaly in the W Pyrenees s.s., the Labourd anomaly, is also attributed to higher-density mantle rocks, and has been interpreted based on seismic tomography, as a continuous rise of the lithospheric mantle related to inversion of the hyperextended passive margin (Fig. 7c, (1))[51]. The decrease in shortening from E to W resulted also in a lower topographic relief and shallower exhumational levels in the W Pyrenees (Fig. 7c). Further to the W, the Pyrenees s.s. transition into the Basque Pyrenees, which grew by inversion of the Basque-Cantabrian Basin and northward subduction of Iberia (Fig. 1d). Shortening was lower than in the Pyrenees s.s. and involved primarily inversion of the extensional structures[45,52]. A very high magnetic and gravimetric anomaly in the vicinity of Bilbao can be explained by mantle and lower crustal rocks near the surface, underlying the sedimentary layers and continuous with the lithospheric mantle (Fig. 7c, (2), section A-A')[52]. The reconstruction of the pre-collisional extensional basins indicates that the Pyrenean and Basque-Cantabrian basins were linked and offset by at least 100 km, while the Parentis basin further N did not link to the Pyrenean basin (Fig. 1d)[45]. Inversion of the extensional system initially produced limited shortening in the Parentis Basin, with deformation primarily localised in the Pyrenean-Basque Cantabrian basins[45], consistent with the model evolution as shown in M2. Despite the intermediate basin offset between the Pyrenean and Basque-Cantabrian basins, which favours opposite polarity subduction, the orogen exhibits the same subduction polarity along strike. Same-polarity subduction with intermediate basin offset requires pre-existing weaknesses (M4, Fig. 6b), which are likely provided by a pervasive N-dipping Variscan pre-extensional fabric[6,45]. The basin offset can nowadays be seen in the kink in the transition between the Pyrenees s.s. and Basque Pyrenees (Fig. 7c, (3)). Higher amounts of shortening and pro-side deformation leading to southward translation of the Pyrenees s.s. likely counteracted the basin offset in the E and explains why the Basque Pyrenees and Pyrenees s.s. roughly form a continuous mountain belt with the basin offset only visible in the kink of the transition zone. The exhumed fragments of mantle rocks on the N-(retro-) side can also be observed in all of our inversion models. Model 4 furthermore shows that in the case of intermediate offset and same-polarity subduction conditions, a larger piece of lower crust and

**Fig. 8 | Diagrams showing the typical evolutionary characteristics of linkage during rift basin inversion.** The evolution is subdivided into a juvenile stage shown as plan-view and a mature stage shown as plan-view with corresponding cross sections. **a** Characteristic evolution of linkage during extension. **b** Characteristics of a mountain belt that evolves into opposite-polarity subduction through linkage of the extensional inner faults. **c** Characteristics of a mountain belt that evolves into same-polarity subduction during inversion of a linked rift basin. Note that it is also possible to link the two outer faults of the linked rift basin (model and diagram not shown). In this case, the resulting opposite polarity in subduction and growth direction would increase the offset between the two rift basins. This scenario is, however, energetically the least favourable and thus not depicted.

lithospheric mantle can be sheared off the retro-plate and become trapped at shallow depth on the retro-side. This mechanism potentially explains the shallow lower lithosphere in the Basque Pyrenees near Bilbao (Fig. 7c, (1)). We speculate that the location of the Labourd anomaly (Fig. 7c, (2)), situated at the W end of the pre-collisional Pyrenean basin, is similarly related to deformation of the basin linkage zone. The Pyrenees are thus a prime example of a mature same-polarity subduction orogen that also exhibits several secondary inversion features and highlights the complexity involved in the inversion of linked extensional basins.

## Discussion

Using coupled geodynamic-landscape evolution models and a force-minimisation analysis, we investigated the 3D aspects of extensional inheritance on mountain building, with a focus on how offset linked rift basins form and influence subsequent orogenesis. For the extensional phase we find that (i) extensional basins with closest proximity and lowest lithospheric strength are most likely to link during rifting, and

(ii) linkage during extension can be subdivided into two phases: During juvenile linkage, the weak zones form local basins that progress towards each other and exhibit a drainage system that mimics the local relief (Fig. 8a). At the mature linkage stage, a connected basin has formed with river flow perpendicular to the basin boundary, enhanced offset on the inner faults and lower offset and topography in the linkage zone (Fig. 8a). Inverting the linked extensional basins leads to mountain building which can similarly be sub-divided into a juvenile and mature phase. (i) In the juvenile stage of basin inversion, sub-duction polarity plays no significant role, and the mountain belt can be described by continuous elevated topography mimicking the shape of the rift basin, and river flow perpendicular to the local strike of the mountain belt topography (Fig. 8a). (ii) With ongoing shortening, the mountain belt structure is determined by the subduction polarity, which depends on basin offset and pre-existing weaknesses. The default mode for the inversion of linked basins with very small offset < ~ 30 km is same-polarity subduction. Once the basin offset is larger than ~ 30 km, the default mode is opposite polarity subduction.

Inherited weaknesses with a preferential orientation that provide a deformational gain will induce same-polarity subduction also if the basin offset is large. Mature orogens with opposite-polarity subduction are characterised by an opposed growth direction of the different orogen branches, removing the structure inherited from the extensional basins. The transition zone between the two subduction directions is characterised by a topographic low (Fig. 8b). On the contrary, orogens with same-polarity subduction are characterised by a uniform growth direction and continuous high topography with river flow perpendicular to the local orogen strike, and retain to first order the inherited linked basin structure (Fig. 8c). These variable characteristics show that the orientation of the rivers and distribution of topography may be used as a first order diagnostic feature to infer the pre-collisional extensional structure and the subduction polarity of mountain belts forming by inversion of linked rift basins.

These evolutionary systematics provide a coherent explanation for the structural characteristics observed in several major inversion orogens, including the Greater Caucasus, Atlas, and Pyrenees. The Greater Caucasus exemplifies a mature same-polarity subduction orogen, in which the inherited extensional template is preserved. The Atlas is a juvenile inversion orogen where subduction polarity does not play a significant role, and highlights that the inherited pre-collisional configuration determines mountain belt structure on different scales. The Pyrenees are a mature same-polarity orogen, in which lateral differences in orogen structure related to rift inheritance are overprinted by differences in the amount of crustal shortening, which is likely one of the major differences when compared to the Greater Caucasus, another mature same-polarity orogen. Observed large high-density bodies at shallow depth in the Pyrenees can furthermore potentially be explained by the inversion of a linked offset rift basin. Our results can be broadened to other oroclines and arcuate orogens that form in locations of previous rift zones, or where the polarity of subduction is unknown or debated. It is for instance debated whether the European Alps contain a subduction polarity reversal between E and W[53,54]. Although our models do not reproduce the complex deformation history of the Alps, our evolutionary framework (Fig. 8) suggests that their continuous high topography, in the absence of a central topographic low, is consistent with same-polarity subduction. Subduction polarity reversal has also been proposed for mountain building in the Hindu Kush – Pamir region[55]. The distribution and propagation-direction of shortening[55], as well as lower topography in the transition zone between both opposite-polarity slabs fits well with the characteristics of an opposite polarity subduction orogen as shown in Fig. 8b. Comparison of model results with natural systems shows that opposite-polarity subduction is rather the exception than the norm, indicating that inherited directional weaknesses play an important role in determining subduction polarity in inversion orogens on Earth.

We conclude that the inversion of linked rift basins, and in particular the resultant subduction polarity, plays a decisive role in mountain belt evolution as part of the Wilson cycle. Juvenile inversion produces continuous high topography, river systems oriented perpendicular to the local orogen strike, and mountain belts that replicate the morphology of the precursor rift basins. Mountain belt structure in mature orogens is largely governed by the subduction polarity and the amount of shortening. Same-polarity subduction retains the extensional template, creates an orogen with continuous high topography, and requires either low basin offsets (a few tens of kilometres) or favourably oriented inherited weaknesses. In contrast, opposite-polarity subduction obliterates the extensional template, reverts the basin configuration, creates a characteristic topographic low in the transition zone, and is the default behaviour for basin-offsets exceeding a few tens of km. Detailed comparison between model results and natural orogens supports the utility of our framework and highlights the critical role of inherited structures in controlling strain

localisation. Our modelling and force-balance analysis provide a simple explanation for the observed variability in subduction polarity among inversion orogens and offer a unified framework for understanding the non-cylindrical nature of many mountain belts on Earth.

## Methods
### Geodynamic model
We use the 3D ALE finite element model pTatin3D[24,25] to solve for incompressible momentum (1), mass (2), and thermal energy conservation (3) in a 3D cartesian box:

$$\nabla \cdot (2\eta\dot{\boldsymbol{\epsilon}}) - \nabla P = \rho\boldsymbol{g}, \tag{1}$$

$$\nabla \cdot \boldsymbol{v} = 0, \tag{2}$$

$$c_p\rho_0\left(\frac{\partial T}{\partial t} + \boldsymbol{v} \cdot \nabla T\right) = \nabla \cdot (k\nabla T) + H + v_z\alpha g_z T\rho, \tag{3}$$

where $\eta$ is the material viscosity, $\boldsymbol{\epsilon}$ is the strain rate tensor, $P$ the pressure, $\rho$ the material density, $\boldsymbol{g}$ the gravitational acceleration vector, $\boldsymbol{v} = (v_x, v_y, v_z)$ is velocity, $c_p$ is the specific heat capacity, $T$ is the temperature, $t$ is time, $k$ is the heat conductivity, $H$ the radiogenic heat production rate. Note that the subscript $z$ indicates the vertical direction. Density varies as a function of temperature

$$\rho(T) = \rho_0(1 - \alpha(T - T_0)), \tag{4}$$

with the reference density $\rho_0$ corresponding to the density at temperature $T_0$. We do not include changes in density as a function of pressure.

Model materials deform either by frictional-plastic deformation or non-linear temperature-dependent viscous creep. Frictional-plasticity is modelled using a Drucker-Prager yield criterion,

$$\tau_Y = C\cos(\phi) + P\sin(\phi), \tag{5}$$

where $\tau_Y$ is the deviatoric yield stress, $\phi$ is the angle of internal friction, and $C$ is material cohesion. Material fails by frictional-plastic deformation when

$$\sigma'_{II} > \tau_Y, \tag{6}$$

where $\sigma'_{II}$ is the square root of the second invariant of the deviatoric stress tensor $\boldsymbol{\sigma}' = 2\eta\dot{\boldsymbol{\epsilon}}$. We track material damage ($\epsilon_{plastic}$) via the second invariant of the accumulated plastic strain. Strain weakening is accounted for by a linear decrease in the angle of friction from 15° to 2° over the pre-defined strain interval $0.5 < \epsilon_{plastic} < 0.7$. Strain weakening, as parameterised here, accounts for the combined effects of permanent material damage occurring during faulting. We do not apply regularisation methods to eliminate the mesh-dependence of the strain weakened material. Tests with different strain weakening thresholds did not show a first-order effect on model results.

Viscous deformation is modelled using a $T$- and $P$- dependent power-law rheology

$$\sigma'_{visc} = f A_c^{\frac{-1}{n_c}} (\dot{\epsilon_{II}})^{\frac{1}{n_c}} \exp\left(\frac{Q + VP}{n_c RT}\right), \tag{7}$$

where $\sigma'_{visc}$ is the square root of the second invariant of the deviatoric stress, $f$ is the scaling factor used to model different geological scenarios, $A_c$ the pre-exponential factor, $n_c$ the power-law exponent, $Q$ the activation energy, $V$ the activation volume, and $R$ the universal gas constant.

## Surface processes model

We use the landscape evolution model FastScape to model landscape evolution above sea level[26,27]. FastScape solves for stream power law erosion ($K_f$-term in eq. (8)), hillslope creep ($K_d$-term), and in-stream sediment deposition (G-term):

$$\frac{\partial h}{\partial t} = U - K_f A^m S^n + K_d \nabla^2 h + \frac{G}{A} \int_A \left( U - \frac{\partial h}{\partial t} \right) dA, \qquad (8)$$

where $h$ is elevation, $t$ is time, $U$ is uplift, $K_f$ is the stream power law coefficient termed fluvial erodibility, $A$ is drainage area, $S$ is slope, $m$, $n$ are the stream power law exponents, $K_d$ is the hillslope diffusion coefficient, and $G$ is the depositional coefficient. Rivers are always connected to one of the four model boundaries through bridging of local minima, ensuring hydraulic connectivity. We use the lake-filling algorithm to fill local minima. The sealevel is set at -500 m with respect to the model sides. Sediments transported to the shoreline are deposited with a mass-conserving aggradation scheme, filling the marine extensional and foreland basins with sediments from the bottom. $m$, $n$ and $G$ are relatively well known, with $G \approx 1$[56], $m/n$ varying from 0.3 to 0.5[57], and $n$ varying from $1 - 3$[58]. Here, we use the typical values $n = 1$, $m = 0.4$, $G = 1$[4]. The fluvial erodibility spans a wide range as it incorporates variations related to e.g., rock strength, climate, or vegetation, with values ranging from at least $1 \times 10^{-6}$ m$^{0.2}$/yr to $1 \times 10^{-4}$ m$^{0.2}$/yr, given $m/n = 0.4$, $n = 1$[57]. The efficiency of erosional versus tectonic flux can be expressed by the non-dimensional Beaumont number $Bm$[4], which, given $m$, $n$ as used here, simplifies to:

$$Bm = \frac{\rho_c g z_{dec}^2 \upsilon_c (1 + G)}{0.5^{0.2} K_f W_{min}^{0.8} F_{int}}, \qquad (9)$$

where $z_{dec} = 25$ km is the depth to the main crustal decoupling horizon, $\rho_c = 2750$ kg/m$^3$ is the average crustal density, $\upsilon_c = 1$ cm/yr is the convergence rate, $W_{min} = 80$ km is a typical length-scale of minimum orogen width, and $F_{int} \approx 2.7 \times 10^{12}$ N/m is approximately equivalent to the maximum integrated crustal strength of undeformed foreland crust above $z_{dec}$ during mountain building as measured in the model (see supplementary material for deviatoric stress plots). We chose $K_f = 0.5 \times 10^{-5}$ m$^{0.2}$/yr for all our models. This set of parameters creates strength-limited mountain belts with Beaumont number $Bm = 3.5$, a value typical for mountain belts on Earth that are not exposed to wet and seasonal climatic conditions. We use a constant hillslope diffusion coefficient of $K_d = 1 \times 10^{-2}$ m$^2$/yr[59].

The landscape evolution model is two-way coupled to the geodynamic model. FastScape has the same amount of nodes as the mechanical Q2-nodes of pTatin3D's finite element mesh. Hence, FastScape truly sits on top of the geodynamic model. After each mechanical timestep, the landscape evolution model is advected according to the computed velocity field at the surface of the geodynamic model. Changes in topography are then computed using eq. (8) and the associated lake-filling and aggradation algorithms for deposition. The new topography is given back to the geodynamic model, and timestepping continues. We note that we limit the FastScape timestep to 1000 yr so that several FastScape timesteps are normally executed per mechanical timestep, which is dynamically computed based on a Courant-Friedrichs-Lewy condition and typically in the order of 9000 yr. Deposited sediments have the same properties as the upper crust, and we track the stratigraphy of deposited sediments in the geodynamic model at 5 Myr intervals.

## Modelling setup

The geodynamic model is laterally uniform and vertically stratified consisting of upper and middle crust to 25 km depth, 10 km of lower crust up to the Moho at 35 km, lithospheric mantle down to a depth of 120 km, and sub-lithospheric upper mantle down to the lower model boundary (Fig. 9). The model has dimensions of 1200 km x 600 km x 600 km and is subdivided into 256 x 128 x 128 finite elements, resulting in a horizontal resolution of 4.7 km per Q2-element. Vertically, the mesh is non-uniform, with a vertical resolution of 2.0 km in the upper 150 km of lithosphere and 8.6 km in the lower 450 km of the model. We base our model materials on a few well-established laboratory-derived flow laws and use the scaling factor $f$ to account for different geological scenarios and uncertainties when extrapolating laboratory-derived flow laws to nature. Specifially, we base model materials on the 'wet' quartz flow law from ref. 60 for sediments and upper and middle crust, the 'dry' Maryland diabase flow law from ref. 61 for lower crust, and the 'wet' olivine flow law from ref. 62 for mantle material. The lithospheric mantle is scaled by a factor $f = 10$ to account for compositionally stronger, e.g., melt-depleted, conditions. The crustal materials based on the 'wet' quartz flow law are scaled by a factor $f = 0.5$. The latter leads to mountain belts with at most 8 km in height, similar to the maximum topography as observed on Earth. Material densities are chosen to be simple and represent typical values of the Phanerozoic lithosphere (Fig. 9 and Supplementary Table 1). The lithospheric mantle is depleted by 15 kg m$^{-3}$ with respect to the underlying sub-lithospheric mantle, which is typical for Phanerozoic lithosphere[63].

The initial temperature distribution in the model domain is laterally uniform and at conductive steady state, with 550 °C at the base of the crust and 1330 °C at the base of the lithosphere, resulting in a surface heat flow of ~53 mW m$^{-2}$ and a mantle mantle heat flow ~21 mW m$^{-2}$, which is typical for Phanerozoic lithosphere[64]. $k$ is linearly increased from $k = 2.25$ W m$^{-1}$K$^{-1}$ to $k = 52.0$ W m$^{-1}$K$^{-1}$ between 1330 °C and 1340 °C, in order to maintain the heat flow at the base of the lithosphere, prevent the model from cooling, and resembling mantle convection at high Nusselt number[65]. This approach leads to an adiabatic gradient of 0.4 °C/km in the sub-lithospheric upper mantle. All the side boundaries are insulated, the bottom boundaries are described by Dirichlet boundary conditions with 0 °C at the top and 1522 °C at the bottom boundary.

To model continental rifting and orogenesis, we apply velocity boundary conditions on the E and W faces of the model; material inflow or outflow on the sides is compensated by outflow/inflow at the bottom boundary. We apply extensional boundary conditions of 0.5 cm/yr at each side, for 12 Myr and then apply shortening with 0.5 cm/yr at each side. The transition from extension to shortening occurs at a linear rate of velocity change between 11.5 Myr and 12.5 Myr of model evolution. The side boundaries have vertical roller conditions, the bottom boundary is characterised by free slip, and the top boundary is a true free surface subject to erosion and deposition by the surface processes model.

Many mountain belts on Earth exhibit to first order a decoupling between crust and mantle at ~20−25 km depth, one-sided subduction of the lithospheric mantle, and mountain building related to crustal thickening[6,38,66−68]. The chosen rheological and thermal setup (see Fig. 9) allows for a decoupled evolution of crust and mantle and represents this first-order observation. It's very typical that mountain belts not only exhibit thick-skinned deformation of crustal-scale thrust sheets, but also thin-skinned thrusting related to weak and shallow décollement horizons in the upper crust, as observed amongst others in the Pyrenees, Alps, or Caucasus. Thin-skinned thrusting changes the structural style, e.g., ref. 18, but is deliberately not part of the model, as the numerical resolution is not high enough to properly resolve this deformational style.

## Model Limitations and extended parameter space

Many pre-collisional extensional basins exhibit a long phase of several 10s to 100 Myr of post-rift cooling before basin inversion. We do not include a phase of post-rift quiescence, but seamlessly transition from extension to shortening in our models. Buiter et al.[16] showed with 2D

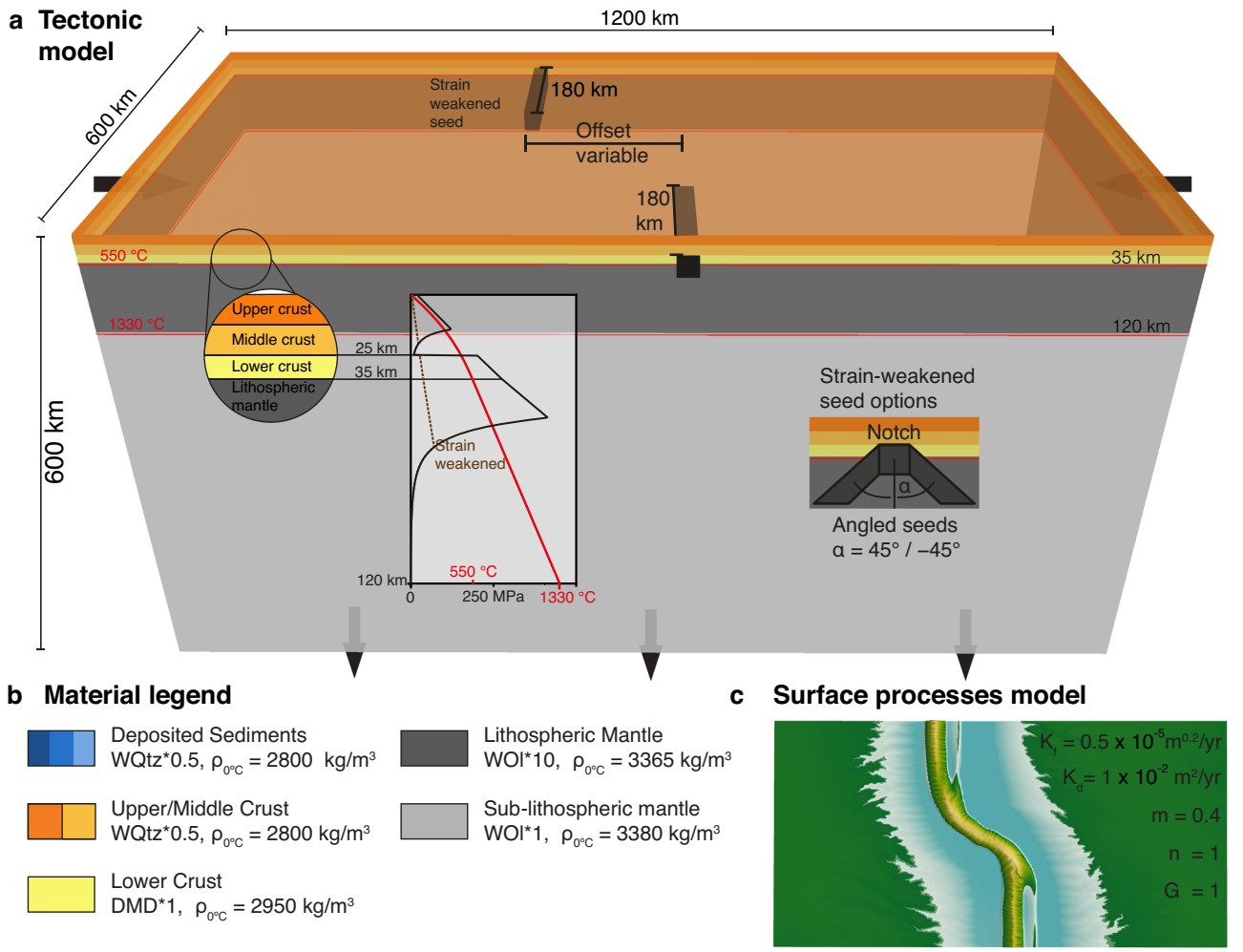

**Fig. 9 | Modelling setup. a** Thermo-mechanical model pTatin3D with initial vertically uniform layering and temperature setup. The zoom on the left shows the different layers of the lithosphere with the associated yield-strength envelope. The inset on the right shows the different options for strain-weakened seeds, either a simple rectangular notch (M1-M3, SM1, SM2), or elongated and angled seeds (M4, SM3). **b** Material legend with flow law and reference density at 0°. Blues of deposited sediments change every 5 Myr. WQtz is the wet quartz flow law based on ref. 60; DMD is the 'dry' Maryland diabase flow law based on ref. 61, WOl is the 'wet' olivine flow law based on ref. 62 (see Supplementary Table 1). **c** Surface processes model FastScape with parameters used in all the models. FastScape is coupled with the tectonic model in a two-way manner.

geodynamic models that varying amounts of post-rift cooling have a very limited effect on the structures developing during inversion, because the weak fault zones formed during extension retain their weakness during post-rift cooling and are therefore reactivated during inversion. We also tested the influence of post-rift cooling for one of our models and found that it has no first-order influence. Furthermore, the force minimisation principles and analysis presented above also hold for geological situations of prolonged post-rift subsidence. We therefore expect that the first-order features observed in our modelling are also applicable to mountain belts exhibiting a phase of post-rift cooling. Further studies should nonetheless investigate the effect of post-rift cooling in detail, including an investigation of transient weakening effects and fault healing mechanisms.

Many mountain belts, like for instance the Alps or Himalaya-Tibet, exhibit a phase of oceanic subduction before continent-continent collision. We do not apply extensional boundary conditions in our models beyond crustal break-up and therefore do not form a large oceanic domain. Future research should focus on this aspect and also include melting processes and the formation of new oceanic lithosphere.

We only apply frictional-plastic softening as a weakening mechanism in our models. Although the reactivation of weak

frictional-plastic shear zones is characteristic of inversion orogens - such as those examined in this study - we note that additional permanent or transient weakening processes may play a role during extension and orogenesis, like grain-size variations, shear heating, or metamorphic phase changes. Future modelling efforts should explore these additional mechanisms. The model results and force balance analysis presented here can serve as a comparative baseline for such investigations.

We only use a uniform surface process efficiency along-strike in the orogen. A strong lateral gradient in the efficiency of erosion, e.g., related to strong climatic gradients or variability in rock erodibility, would induce a higher widening rate in areas with lower erosional efficiency, and vice versa[4]. In this case, the along-strike differences in surface processes would potentially overprint the mountain belt structure inherited from a previous extensional event, and the previous basin structure is likely only visible in the transfer zone, i.e., in a laterally confined area. A similar effect can be obtained by varying the amount of convergence along strike, as discussed for the Pyrenees. Furthermore, along-strike differences in the strength of the colliding plates would have a similar effect. Higher plate strength would lead to a higher and narrower mountain belt, and vice versa[4].

We investigated the simplified case of inverting two offset rift basins. In natural systems, however, multiple offset basins are likely to occur. Based on our model results and force-balance analysis, we predict that inverting such systems would behave as if composed of several basin pairs similar to those examined here, potentially involving multiple subduction polarity flips in the absence of dominant inherited weaknesses guiding deformation. Future research should explore this topic.

## Data availability

All data supporting the findings of this study are contained within the article and Supplementary Information. Animations of the models presented in this study can be found in the supplementary material.

## Code availability

Numerical models are computed with published methods and codes. The 3D thermo-mechanical tectonic model can be accessed via https://bitbucket.org/ptatin/ptatin3d. The surface processes FastScape can be accessed via https://fastscape.org/; specifically, we use fastscapelib-fortran located at https://github.com/sebastianwolf/fastscapelib-fortran.

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

## Acknowledgements

We want to thank Thomas Theunissen and Edoseghe Osagiede for inspiring discussions. We acknowledge Sigma2 for HPC on server Betzy through project nn4704k. We acknowledge the support of the research project Structure and Deformation of Salt-bearing Rifted Margins (SABREM), PID2020-117598GB-I00, funded by MCIN/ AEI /10.13039/ 501100011033.

## Author contributions

Author contributions following CRediT: S.G.W.: Conceptualisation, Methodology, Software, Validation, Formal analysis, Investigation, Data

Curation, Writing - Original Draft, Writing - Review & Editing, Visualisation. R.S.H.: Conceptualisation, Investigation, Resources, Writing - Review & Editing, Supervision, Project administration, Funding acquisition. J.A.M.: Investigation, Writing - Review & Editing. D.A.M.: Methodology, Software, Writing - Review & Editing.

## Funding

## Competing interests
The authors declare no competing interests.
