## [Transparent Peer Review file · Nature Communications]

Rift linkage and inheritance determine collisional mountain belt evolution

Corresponding Author: Dr Sebastian Wolf

Version 0:

Reviewer comments:

Reviewer #1

(Remarks to the Author)

The paper presents new, state-of-the-art numerical scenarios that combine a 3D thermomechanical model with a landscape evolution model to simulate the influence of rifting geometry on the evolution of a collisional mountain belt. The authors explore scenarios with different offsets between two rift branches, demonstrating that the orogenic structure is controlled by the rifting geometry. Additionally, they present an analytical approach that offers a convincing physical explanation for the numerical results, highlighting the conditions under which rift branches with different offsets may connect.

I recommend publication of the article after the following points are addressed:

In scenario M2, which involves a large offset of the weak seed (400 km), the simulation evolves into one-sided subduction with an essentially two-dimensional (cylindrical) geometry. As described by the authors, this occurs because the eastern rift branch is aborted, and the western basin propagates southward, resulting in a linear rift geometry. However, this result likely depends on the size and rheology of the weak seeds: if both branches contain large, sufficiently weak seeds, rifting might develop on both the western and eastern sides, potentially leading to a successful rifting scenario with a large offset. It is probable that the small size of the weak seed contributed to the failure of one branch, making the resulting scenario redundant with scenario M1.

In scenario M3, which uses an intermediate offset of 200 km between weak seeds, the simulation produces an interesting subduction pattern with opposite polarity, resulting in a complex, three-dimensional orogenic system (Figure 4c). My question concerns the robustness of this result: is opposite-polarity subduction a systematic outcome for intermediate-offset (~200 km) scenarios with simple weak seeds, or is it a fortuitous result? Given that the weak seeds in these scenarios are symmetric in East–West cross-sections, and that 2D thermomechanical models with similar seeds do not allow for a priori prediction of subduction polarity (since small perturbations in the thermal and compositional structure of the lithosphere can change the polarity) the observed pattern may not be systematic. Additionally, the numerical setup explores only a single offset per scenario. What would be expected if multiple branches with intermediate offsets (~200 km) were present in the domain? As the authors convincingly argue, polarity is sensitive to inherited geometries of the weak zones. Therefore, I suggest that the offset plays only a secondary (or negligible) role in controlling subduction polarity.

Minor points:

The authors use cardinal points to indicate features in the numerical scenarios. This is a helpful approach, especially for 3D models with complex geometries, as it facilitates understanding of the numerical results. However, since the scenarios are generic and not intended to represent any specific orogenic system on Earth, it is important to define the direction of the initial extension. This would improve the reader's comprehension of the results. I suggest adding a clarification in lines 79–81, such as: "We assumed the direction of extension to be East–West." Although this is mentioned in the Methods section and indicated in Figures 2–5, including it in the main text would enhance clarity.

Continuing on this topic, the description of the Model 3 results is unclear. The authors state in lines 119–120 that "the right fault in the northern basin and the left fault in the southern basin record more offset and therefore also slightly higher rift-flank topography." However, the combination of cardinal points with terms such as "right" and "left" is confusing. Please consider replacing these with "eastern" and "western" for clarity.

Model names appear in both lowercase (e.g., line 109) and uppercase (e.g., lines 221–222). Please standardize their formatting throughout the manuscript.

Reference numbers in lines 461–469 should be placed within square brackets [] to avoid confusion with exponents.

Reviewer #2

(Remarks to the Author)

Review of Wolfe, S.G., Huismans, R.S., Munoz, J.A., May, D.A.

Rift Linkage and inheritance determine collisional mountain belt evolution

For Nature communications

18 July 2025

This paper presents the results of a series of sophisticated 3D thermomechanical simulations of lithospheric extension followed by shortening. The objective is to consider how offset or linked rift segments affect subsequent orogen growth. The thermomechanical software is, in addition, coupled with a 3D surface processes modelling package that simulates erosion from growing relief and deposition of sediments in developing basins. 3D models simulate orthogonal, segmented rifting that is allowed to develop to the point of continental breakup followed by convergence to juvenile (or partial) inversion, mature (or complete) inversion and beyond. Extensional and compressional stresses are parallel. The simulations consider the effect of offset distance between rift segments on orogen evolution. The authors conclude that this factor influences initial growth of the mountain belt being strongly impacted by the offset distance of rift segments. Later orogen growth is governed by subduction polarity, which itself is mainly dictated by the asymmetry of the main conjugate rift faults. The model input parameters and uncertainties are described at the end of the paper. Simulation results are compared to three mountain chains, the Causases, Atlas and Pyrenees, which are in turn interpreted in terms of rift geometry inheritance.

This is an excellent paper from one of the leading thermomechanical modelling groups in geosciences. The methodology is sound and well established with the innovative combination of two softwares to provide a more complete 3D earth system simulation. It is well written and illustrated with impressive videos of model development. The authors state clearly the uncertainties and limitations of their models. The conclusions and comparisons with natural examples are valid, even if some queries need addressing. The results will provoke much discussion and interest amongst a wide readership interested in Wilson cycle lithospheric processes. It is also a nice demonstration of cutting edge multidimensional thermomechanical modelling combined with surface processes modelling. I feel that the surface processes aspects of the modelling do not get the attention that perhaps they merit. However, this may perhaps be the subject of a later paper. I recommend this paper for publication once the points below and other minor comments on the pdf are addressed.

Mary Ford,

Université de Lorraine, Nancy.

Lines 53_55: Some analogue modelling studies do actually address these issues. What is missing from the majority of analogue models, it seems to me, is a full crustal/lithospheric treatment of these issues.

79: Can you explain why this rate and period were chosen? This appears to thin the lithosphere to the point of mantle exhumation and breakup?

81: State how long the shortening phases were and what the rate of shortening was. The duration of shortening appears to be variable (roughly 44 Myrs to 63 M Yrs) but always a lot longer than 12 Ma. Give the reasoning used for the duration and rate of the shortening ? Attaining a set amount of shortening for example ?

95: southern?

96: southern? Why use left and right? Why not stick to the cardinal directions of the compass as on Figures 2-5 of models and elsewhere in text?

96: What do you mean by "Mature"? Fully or completely inverted? Maybe explain what you mean here and/or give reference. Same point for Juvenile inversion.

119 – western fault

125-6 : Oppositely dipping subduction - how do these link ? Can you give some detail here of how this works in the models?

129 : not clear here. Reverting is not the appropriate word as it means to return to a former state. Inverting or switching would perhaps be clearer.

178 : But you say you have already assumed this in line 166.

212 But there is a (large) offset in Model 2

273-275 : Incomplete sentences... punctuation problem?

288 : EHA is not defined. Also this sentence is not clear and needs a rewrite.

295 : Compare to what stage of these models ? You could define this in the videos

304 : Its not clear how do the Atlas mountain belt highlights this point that basin linkage and associated changes in mountain belt structure occur at different scales during inversion.

310 : The Mauleon basin is a small (but important) depocentre within the western Pyrenean system (SW of Pau). It is misleading and inappropriate here to refer to the whole Pyrenean rift (ss) by this name.

310 'rifted margin'. Is the Pyrenean depocentre a rift or a rifted margin? So far you have been talking about rifts. If the latter then which margin are your referring to?

310 : Suggest propagated would be clearer than 'progressed'.

313 Add « decreasing from east to west » at end of sentence.

317 : Suggest rewording : « Exhumed fragments of mantle rocks on the N-flank (retrowedge) of the orogen, are found embedded in metamorphosed Mesozoic sedimentary cover between inverted crustal extensional blocks ». I suggest citing not just Teixell et al. (2018) but also, for example, Angrand et al. (2020, J. Geol Soc London) and/or Asti et al. (2022 ; Earth Science Reviews)

320 : In model of Wang et al. the rise of the mantle lithosphere occurred during hyper-extension. This configuration was essentially preserved due to very low convergence in this area of the westernmost north Pyrenees.

393 : I don't think this explains the presence of large high density bodies. The offset between the Basque Pyrenees and the main Pyrenees ss can certainly be explained by inversion of a linked offset rift basin. But the presence of high density bodies is surely related to exhumation of deep lithosphere during hyper-extension and their subsequent incorporation into the orogen. In other words their presence is related primarily to the initial style and amount of extension in the rift system...

409 : and the amount of shortening?

411-414 : Is there any natural mountain belt that demonstrate this behaviour?

481 : This section and Figure 9 would be better in my opinion early in the paper to allow the reader to understand the model results. For example, the colours codes for results figures are given on figure 9.

Figures

Figure 1. I have located the actual Mauleon basin on figure 1d. As this is a well known and defined local basin, its name is not appropriate for the whole Pyrenean rift system.

Figures 2 to 5. It would be helpful to put the compass points on at least the first model image. Need a key or labels for colours on the cross sections.

Can you separate the three cross sections as complex details are not clear in the areas where they overlap ? Same comment for all cross sections.

Given that the 3D map views are labelled a, b, c, it would probably be better not to use A, B, C for the cross sections ?

Reviewer #3

(Remarks to the Author)

The manuscript titled "Rift linkage and inheritance determine collisional mountain belt evolution" by Wolf and co-authors presents 3D numerical models of lithosphere extension directly followed by convergence. The main aims of the modelling are to better understand linkage of rift segments during extension and the impact of the rift geometry on the subsequent convergence, particularly the subduction polarity. The applied numerical model consists of a lithosphere deformation model coupled to a landscape evolution model. The study includes a very interesting application of force analysis to explain linkage of rift segments and the subduction polarity during subsequent convergence and rift inversion. Furthermore, the study discusses the application of the model results to the Greater Caucasus, Atlas and Pyrenees. There are still many open questions concerning the impact of 3D features on rifting and rift inversion and the manuscript, hence, presents a very interesting study on the application of deterministic mathematical models to better understand these geodynamic processes. The presented results demonstrate the importance of the pre-orogenic 3D rift geometry on the subsequent orogeny related to the inversion of the rift basins. The authors apply a force analysis to explain the 3D model results which is very useful for interpreting and understanding the numerical results. I find the study of interest for a wide readership in Geosciences and, hence, find the study suitable for publication. I have several comments, with the main aim to make the model results more transparent, which the authors could consider.

The applied lithosphere deformation model represents an end-member scenario in which the deformation of the crust and lithospheric mantle is dominated by brittle-frictional processes. The maximal friction angles of both crust and mantle lithosphere are only 15 degrees (providing a deeper brittle-ductile transition compared to a friction angle of 30 degree) and only strain weakening due to a reduction of the friction angle with strain is considered which furthermore generates a permanent weakening. No other weakening mechanisms are considered, such as due to grain size variation or shear heating, and no transient weakening mechanisms are considered which would allow for healing and re-strengthening of shear zones and faults that were active during rifting. Since weakening mechanisms other than friction-weakening and healing most likely occur in nature, it is not clear how representative and robust (with respect to model variations by including other weakening mechanisms and healing) the presented results are. Ideally, also another end-member model would be presented with different weakening mechanisms and non-permanent weakening to test how dependent the first order results are on the particular model assumptions. I can imagine that such additional models are outside the scope of the study, but I maybe the authors can at least explain the particularities of their model and discuss the potential impact of different weakening mechanisms and healing on their results.

For the force analysis, it would be useful to also know the values of the depth-integrated strengths, for example F_{weak} and F_{strong} (in N/m ; discussed in the supplementary information) and how they vary laterally because these values could then be compared to other estimates for these strengths, such as plate driving forces (e.g. slab pull, ridge push or force balance at continental plateaus). It would be also useful to have figures that show the cross section of the models with the corresponding deviatoric stress field to be able to see the stress magnitudes and ideally the brittle-to-ductile transition in the models. Similarly, it would be useful to have additional figures of the cross sections that indicate with a colormap which regions deform in the brittle-plastic regime and which ones in the viscous regime. This can further help to compare the model results with field observations or earthquake observations. Such figures could be put in the supplementary material. Currently, there are no figures that show any stress fields, although they are the fundamental values used in calculating the forces, and it is not clear how the brittle-frictional and viscous deformations are distributed during orogeny.

Some minor comments and questions:

Line 75 : Please provide the formula for the B_m number. Also, please provide the values used in the simulations; $B_m > 1$ is vague.

Does the landscape evolution model has a significant impact on the collisional model evolution or would a simpler erosion

model yield similar results?

2.2.1. : As mentioned, it would be useful to know also the values of the vertically integrated strengths to compare them with plate driving forces (in N/m).

2.2.2. : The model assumes significant permanent weakening (minimum friction angle of 2 degree in both crust and mantle) and no healing. Hence, it is expected that in the presented models the permanent weak zones developed during rifting control the subsequent collision stage.

Equation (3) : It seems that shear heating due to dissipative deformation (friction and viscous shearing) is not considered? If associated shear heating by dissipative deformation is not considered in equation (3), then energy is not balanced in the model.

Equation (4) : The model is 600 km deep. Why density is not also a function of pressure?

Line 439 : Strain weakening by reducing the friction angle is usually mesh-dependent if no regularization method is applied. How is the mesh-dependence handled in the numerical model?

Line 466 : Please provide the formula for the Bm number.

Line 505 to 509. When the lithosphere is subducted into the sub-lithospheric upper mantle, how can this parameterization of k then produce a correct temperature field around the subducting lithosphere? This parameterization may work well during the extension phase, but how can it be justified for the subduction process?

Line 534 to 537 : This limited effect is controlled by the model assumptions because only a frictional strain weakening mechanism is considered which causes permanent weakening. No transient weakening mechanisms are considered which would allow healing and re-strengthening rift-related faults and shear zones. Therefore, the limited impact of a cooling phase is model related, but not related to natural processes which can involve healing mechanisms.

Best regards,
Stefan Schmalholz

Version 1:

Reviewer comments:

Reviewer #1

(Remarks to the Author)

The authors have adequately addressed all of my comments and concerns in their revised manuscript. I find the current version suitable for publication and recommend acceptance in its present form.

Reviewer #2

(Remarks to the Author)

I have read through the revised manuscript and the reply of authors to reviewers. The authors have carefully addressed and discussed the comments and revisions suggested by reviewers. The paper is now much improved in clarity and represents an important and illuminating contribution to our understanding of rift inversion and orogenesis at a lithospheric scale.

One minor correction needs to be made on figure 1d - change Mauleon Basin to Pyrenean Basin as revised in the text.

I congratulate the authors on this very nice work.

I recommend publication

Mary Ford, universit  de Lorraine, Nancy, France.

Reviewer #3

(Remarks to the Author)

The authors have addressed my comments and clarified the text with respect to model assumptions. They also added new panels in the figures in the supplementary material showing the deviatoric stress which I find very useful. Only in case the authors will make another revision (which is not necessary), I have few minor comments to the manuscript. I also have two comments to their rebuttal.

Minor comments

Line 377: "...if their offset is not too large..."; maybe the authors could quantify what is "not too large" with a dimensionless number, for example, if the ratio of offset to crustal thickness is less than approximately...

Line 451: If H is only the radiogenic heat production rate, please specify this.

Line 510: For completeness, the authors could also specify the value or criteria for the mechanical time step.

Comments to rebuttal and new text in supplementary material

Lithostatic pressure and force balance analysis: It is still not clear to me why the authors assume lithostatic pressure and a constant strain rate for the force calculation. Why don't the authors not use the values of deviatoric stresses calculated in their 3D thermo-mechanical model to calculate F_{int} ? Concerning deviations from lithostatic pressure: figures S1 to S4 all indicate deviatoric stresses of several hundreds of MPa up to 1 GPa according to the colorbar. Since tectonic overpressure magnitudes are similar to deviatoric stress magnitudes, overpressure should be significant in the models. Therefore, if there

are significant deviatoric stresses, then the assumption of lithostatic pressure is not justified. In the rebuttal, the authors say that tectonic overpressure stems from gravitational potential energy (GPE) variations. However, this applies only to situations without far-field deformation. In a lithosphere under compression/shortening, there can be significant tectonic overpressure also without GPE variations if deviatoric stresses are large, as is the case in the presented models. For the yield strength envelope for F_{strong} , shown in figure 6, there is significant overpressure, for example, in the brittle regions of the mantle lithosphere. This has been shown, for example, in the study of Petrini & Podladchikov, *Journal of metamorphic Geology*, 2000.

Response concerning line 439: "We note that this parametrisation is nonetheless commonly used throughout the community and reported in the literature.". This is not a convincing reply, because the fact that others use a certain method does not necessarily mean that the method is correct. I am also not sure which community is meant. The mesh-dependence can affect the calculated load-bearing capacity of a strain-softening material and hence can affect the force calculation. But since the authors do anyway not use their 3D model results, involving strain softening, to calculate the forces, but a simplified 1D yield strength envelope, their force analysis is not affected by the mesh-dependence.

The overall results of the force balance analysis make sense, but the accuracy and numerical robustness of the calculated values of F_{weak} and F_{strong} (in units of N/m) and the resulting forces (in units N) is not clear to me.

Best regards,
Stefan Schmalholz

Response to Reviewers

Sebastian G. Wolf, Ritske S. Huismans, Josep Anton Muñoz, Dave A. May

This document concerns our revision of the paper entitled *Rift linkage and inheritance determine collisional mountain belt evolution*. We thank all three reviewers for very positive, constructive, and useful comments that helped improve our manuscript. We accommodated all changes suggested by the reviewers. Below follows a point-by-point reply to the reviewer comments. Comments are in black font with serif; our answer is written in blue font without serif. Lighter blue text without serif highlights changes to the manuscript text.

Reviewer #1

The paper presents new, state-of-the-art numerical scenarios that combine a 3D thermomechanical model with a landscape evolution model to simulate the influence of rifting geometry on the evolution of a collisional mountain belt. The authors explore scenarios with different offsets between two rift branches, demonstrating that the orogenic structure is controlled by the rifting geometry. Additionally, they present an analytical approach that offers a convincing physical explanation for the numerical results, highlighting the conditions under which rift branches with different offsets may connect.

I recommend publication of the article after the following points are addressed:

We thank the reviewer for very positive, useful and constructive comments that helped improve our manuscript.

Comment 1: In scenario M2, which involves a large offset of the weak seed (400 km), the simulation evolves into one-sided subduction with an essentially two-dimensional (cylindrical) geometry. As described by the authors, this occurs because the eastern rift branch is aborted, and the western basin propagates southward, resulting in a linear rift geometry. However, this result likely depends on the size and rheology of the weak seeds: if both branches contain large, sufficiently weak seeds, rifting might develop on both the western and eastern sides, potentially leading to a successful rifting scenario with a large offset. It is probable that the small size of the weak seed contributed to the failure of one branch, making the resulting scenario redundant with scenario M1.

We agree with the comment by the reviewer. As shown by the models and the force balance analysis, a linked rift forms if the force required to deform the connection between the two offset rift basins (D' ; Fig. 6a) plus the force required to deform the basin with length (D_{seed}) that is being linked to is smaller than the force required to link to the model side (D_b). If there would be weaker and possibly larger seeds, that reduce the integrated strength in the offset basins to be smaller than F_{weak} , then the force required to deform D_{seed} would be smaller and linkage could occur with larger basin offsets. The comment from the reviewer is therefore fully consistent with our analysis and modelling results. To clarify, we added the following sentence to the supplementary material of the manuscript:

This relationship also highlights that if F_{weak} was smaller, linkage would occur at larger basin offsets. A smaller F_{weak} could be related to very weak inherited weaknesses, and could be modelled with weaker and larger initial weak seeds.

We note though that we prefer to use a simple model setup with relatively small weak seeds that are fully strain weakened. This setup prevents over-constraining the models and provides the biggest amount of “freedom” for the models to develop. We do not believe that running more models with artificially even weaker and bigger seeds provides more insight beyond the produced modelling results.

Comment 2a: In scenario M3, which uses an intermediate offset of 200 km between weak seeds, the simulation produces an interesting subduction pattern with opposite polarity, resulting in a complex, three-dimensional orogenic system (Figure 4c). My question concerns the robustness of this result: is opposite-polarity subduction a systematic outcome for intermediate-offset (~200 km) scenarios with simple weak seeds, or is it a fortuitous result? Given that the weak seeds in these scenarios are symmetric in East–West cross-sections, and that 2D thermomechanical models with similar seeds do not allow for a priori prediction of subduction polarity (since small perturbations in the thermal and compositional structure of the lithosphere can change the polarity) the observed pattern may not be systematic.

We thank the reviewer for this comment, that we deliberately split into two for simplicity (Comment 2a and Comment 2b below).

Yes, this scenario is systematic and not fortuitous. We tried different amounts of extension and different amounts of offset, but offsets > ~30 km consistently lead to opposite polarity subduction. Only if there is an existing directional weakness that provides a gain so that the longer same-polarity deformation path is taken, the system would develop into same-polarity subduction (M4). The force balance analysis quantifies this behaviour and provides an explanation why the polarity flip is happening. The nature of the opposite polarity subduction is one of the interesting key outcomes of this study. To highlight that this result is systematic, we added another model with offset 100 km (SM2) to the supplementary material, that also develops opposite polarity subduction, as expected.

We added the following sentences to the results section:

SM2, with an intermediate offset of 100 km between simple weak seeds exhibits very similar behaviour to M3, with opposite polarity subduction as a result of inner fault-linkage. Additional models with similar offsets all develop into opposite polarity subduction. These results show that opposite polarity subduction in intermediate-offset simple seed models is a systematic and repeatable model result.

We refer the reviewer to section 2.2.2, the force balance analysis, that gives a physical explanation for subduction polarity showing that the model results are not only consistent but also quantifiable.

And yes, we also made the same experience that a simple weak seed does not allow for a priori prediction of subduction polarity in 2D models; these models behave like a straight rift system without offset (Model 1). However, as our study shows, when considering small offsets and 3D linkage, subduction polarity is a priori predictable. We think that this part is a very nice result of this study as it provides, to our knowledge for

the first time, a quantifiable understanding for subduction polarity development following rift basin inversion. On a side note, from a modellers perspective who has been frustrated with the a priori unpredictability of 2D models, this result is especially satisfying as it provides physical reasoning. We believe that this topic does not need additional textual changes as it is covered by section 2.2.2.

Comment 2b: Additionally, the numerical setup explores only a single offset per scenario. What would be expected if multiple branches with intermediate offsets (~200 km) were present in the domain? As the authors convincingly argue, polarity is sensitive to inherited geometries of the weak zones. Therefore, I suggest that the offset plays only a secondary (or negligible) role in controlling subduction polarity.

The reviewer raises an interesting question about what would happen with several offset weaknesses. In this case we can only speculate and use simple logic based on the force balance analysis. Deformation will always choose the path of least resistance. Assuming that there are no other inherited weak zone geometries that govern subduction polarity, we would expect these systems to behave as if composed of several basin pairs similar to those examined here. We added the following text to the limitations section of this manuscript:

“We investigated the simplified case of inverting two offset rift basins. In natural systems, however, multiple offset basins are likely to occur. Based on our model results and force-balance analysis, we predict that inverting such systems would behave as if composed of several basin pairs similar to those examined here, potentially involving multiple subduction polarity flips in the absence of dominant inherited weaknesses guiding deformation. Future research should explore this topic.”

We agree that polarity is sensitive to inherited geometries of the weak zones, as shown by the models and analysis. The physical intuition of the reviewer might be correct that in mountain belts on Earth inherited weaknesses play a dominant role in determining subduction polarity – otherwise we would likely see a subduction polarity switch in most orogens. Our study quantifies this process and shows when subduction polarity switches, how structures and topography evolve in response to the polarity of subduction, and how weak inherited weakness need to be to change subduction polarity. We think this is new and a useful contribution. To highlight the point made by the reviewer, we added the following sentence to the second to last paragraph of the manuscript:

Comparison of model results with natural systems shows that opposite-polarity subduction is rather the exception than the norm, indicating that inherited directional weaknesses play an important role in determining subduction polarity in inversion orogens on Earth.

Minor points:

The authors use cardinal points to indicate features in the numerical scenarios. This is a helpful approach, especially for 3D models with complex geometries, as it facilitates understanding of the numerical results. However, since the scenarios are generic and not intended to represent any specific orogenic system on Earth, it is important to define the

direction of the initial extension. This would improve the reader's comprehension of the results. I suggest adding a clarification in lines 79–81, such as: "We assumed the direction of extension to be East–West." Although this is mentioned in the Methods section and indicated in Figures 2–5, including it in the main text would enhance clarity.

We thank the reviewer for this comment and agree. We update the sentence in lines 79–81 as follows (addition in bold font):

*Each model is extended for 12 Myr with a rate of 1 cm/yr in **East-West direction**, followed by a reversal of the velocity boundary conditions and shortening with the same rate.*

Continuing on this topic, the description of the Model 3 results is unclear. The authors state in lines 119–120 that "the right fault in the northern basin and the left fault in the southern basin record more offset and therefore also slightly higher rift-flank topography." However, the combination of cardinal points with terms such as "right" and "left" is confusing. Please consider replacing these with "eastern" and "western" for clarity.

We agree and changed all "right" and "left" to "west(ern)" and "east(ern)".

Model names appear in both lowercase (e.g., line 109) and uppercase (e.g., lines 221–222). Please standardize their formatting throughout the manuscript.

We thank the reviewer for very careful reading! We agree and changed the inconsistent cases. Now the models are named either **Model 1** or **M1** etc.

Reference numbers in lines 461–469 should be placed within square brackets [] to avoid confusion with exponents.

We agree and wrapped these citations with square brackets.

Reviewer #2:

This paper presents the results of a series of sophisticated 3D thermomechanical simulations of lithospheric extension followed by shortening. The objective is to consider how offset or linked rift segments affect subsequent orogen growth. The thermomechanical software is, in addition, coupled with a 3D surface processes modelling package that simulates erosion from growing relief and deposition of sediments in developing basins. 3D models simulate orthogonal, segmented rifting that is allowed to develop to the point of continental breakup followed by convergence to juvenile (or partial) inversion, mature (or complete) inversion and beyond. Extensional and compressional stresses are parallel. The simulations consider the effect of offset distance between rift segments on orogen evolution. The authors conclude that this factor influences initial growth of the mountain belt being strongly impacted by the offset distance of rift segments. Later orogen growth is governed by subduction polarity, which itself is mainly dictated by the asymmetry of the main conjugate rift faults. The model input parameters and uncertainties are described at the end of the paper. Simulation results are compared to three mountain chains, the Causases, Atlas and Pyrenees, which are in turn interpreted in terms of rift geometry inheritance.

This is an excellent paper from one of the leading thermomechanical modelling groups in geosciences. The methodology is sound and well established with the innovative combination

of two softwares to provide a more complete 3D earth system simulation. It is well written and illustrated with impressive videos of model development. The authors state clearly the uncertainties and limitations of their models. The conclusions and comparisons with natural examples are valid, even if some queries need addressing. The results will provoke much discussion and interest amongst a wide readership interested in Wilson cycle lithospheric processes. It is also a nice demonstration of cutting edge multidimensional thermomechanical modelling combined with surface processes modelling. I feel that the surface processes aspects of the modelling do not get the attention that perhaps they merit. However, this may perhaps be the subject of a later paper. I recommend this paper for publication once the points below and other minor comments on the pdf are addressed.

Mary Ford,

Université de Lorraine, Nancy.

We thank the reviewer Mary Ford for a very useful and constructive review and very precise comments that helped improve the manuscript.

Concerning the surface processes aspects: We agree that the surface processes part of the article is very exciting and should be followed up with future work.

We thank the reviewer for providing comments directly in the manuscript PDF and for also exporting these comments in the list below. Some minor textual comments that required only very small changes to the text are not listed below. We have implemented all of these minor textual suggestions. In case we did not follow them, they are listed below.

Lines 53_55: Some analogue modelling studies do actually address these issues. What is missing from the majority of analogue models, it seems to me, is a full crustal/lithospheric treatment of these issues.

We agree with the reviewer. We already write that analogue models only focus on crustal scale in the sentences before. To be even clearer and specific, we changed the respective sentence to:

Hence, the importance of a 3D rift setting with stepping, offset, and linked rift basins on inversion tectonics remains unknown when including a consistent upper mantle-scale perspective.

79: Can you explain why this rate and period were chosen? This appears to thin the lithosphere to the point of mantle exhumation and breakup?

We thank the reviewer for this comment. The rate was chosen because it represents typical plate velocities on Earth; and yes, the period was chosen to create full crustal break-up. We tested shorter periods as well, which had no significant effect. We added the following sentence:

This period of extension leads to full crustal break-up; the applied rate of extension/shortening represents typical plate velocities on Earth.

81: State how long the shortening phases were and what the rate of shortening was. The duration of shortening appears to be variable (roughly 44 Myrs to 63 M Yrs) but always a lot

longer than 12 Ma. Give the reasoning used for the duration and rate of the shortening ?
Attaining a set amount of shortening for example ?

The rate was again chosen to represent typical values of plate movement on Earth; the period was chosen to create at most an intermediate-size orogen larger than the European Alps. The rate is accounted for by the previous comment; we added the following sentence, also addressing model duration:

Models run for 63 Myr but were stopped when the evolving orogen reached the Eastern or Western model side. This period and rate of shortening creates orogens reaching intermediate size, slightly larger than, for instance, the European Alps.

95: southern?w

We agree and changed to *western*

96: southern? Why use left and right? Why not stick to the cardinal directions of the compass as on Figures 2-5 of models and elsewhere in text?

We agree and changed all “left” and “right” to cardinal directions throughout the manuscript.

96: What do you mean by "Mature"? Fully or completely inverted? Maybe explain what you mean here and/or give reference. Same point for Juvenile inversion.

We thank the reviewer for detailed reading, agree, and added the following explanation to the end of the first results paragraph:

Throughout the manuscript, we refer to ‘juvenile’ and ‘mature’ rifts or orogens. Juvenile rifts contain unconnected sub-basins, unlike mature rifts where these are linked (Fig. 2a). Juvenile orogens lack clear subduction polarity and are only composed of inverted extensional structures (Fig. 2b), whereas mature orogens exhibit distinct lithospheric subduction and additional crustal shortening (Fig. 2c).

119 – western fault

We agree and changed accordingly.

125-6 : Oppositely dipping subduction - how do these link ? Can you give some detail here of how this works in the models?

We observe that the linkage between the oppositely dipping slabs is accommodated by successive separation of discrete pieces of lower crust and lithospheric mantle in the transitional domain stemming alternately from both slabs (see Fig. 4c). We describe these structures and the associated process two sentences later. To clarify, we changed the respective text to:

Interaction of the two subducting slabs is characterised by several large pieces of lower crust and lithospheric mantle sheared off from the opposite-polarity subducting lithospheres, and creating a topographic low in the transfer zone (Fig. 4c).

129 : not clear here. Reverting is not the appropriate word as it means to return to a former state. Inverting or switching would perhaps be clearer.

We agree. We think that *reversing* is the right term, as the plan-view shape is reversed. We changed accordingly.

178 : But you say you have already assumed this in line 166.

We agree and changed to:

To summarize, model behaviour can be successfully quantified by assuming that deformation follows the path of least resistance, such that the integral of the depth-integrated lithospheric strength along any given path is minimised.

212 But there is a (large) offset in Model 2

Yes, we agree. We changed this sentence to:

These force minimisation principles also explain same-polarity subduction in cases of a straight rift system without significant offset, as in Models 1 and 2.

227: provides.

We think provide without 's' is correct, as provide refers to inherited weaknesses

273-275 : Incomplete sentences... punctuation problem?

We agree and removed the subsentence:

, and uplift continuing today.

288 : EHA is not defined. Also this sentence is not clear and needs a rewrite.

It was defined in the sentence before. However, we rewrote the respective sentences to:

Looking at the large-scale trends of the Atlas, we find that the Western branch, the so-called High Atlas, trends SW-NE, with a change in strike at the Eastern High Atlas (EHA) towards W-E, which changes again after ~300 km to a SW-NE strike. The northern branch termed Middle Atlas (MA) has yet another orientation with a more Northerly direction (Fig. 7b).

295 : Compare to what stage of these models ? You could define this in the videos

We agree, the sentence now reads as follows:

The mild inversion of extensional basins with low amounts of shortening and incipient subduction indicate that the Atlas is a juvenile inversion orogen where our models show that subduction polarity does not yet play a significant role, as shown in Models 3 and 4 after ~20 Myr of model evolution (Fig. 4b and Fig. 5b)

We think that this is clear definition. We did not label this stage in the videos so as to not create the wrong impression that a specific model represents a specific orogen.

304 : Its not clear how do the Atlas mountain belt highlights this point that basin linkage and associated changes in mountain belt structure occur at different scales during inversion.

We thank the reviewer for detailed reading and specified to:

The Atlas is a fitting example of a juvenile inversion orogen where subduction polarity has no primary influence, and highlights that basin linkage and associated changes in mountain belt structure during inversion occur on different spatial scales, from small-scale steps (10s of km) to large-scale branching (100s of km).

310 : The Mauleon basin is a small (but important) depocentre within the western Pyrenean system (SW of Pau). It is misleading and inappropriate here to refer to the whole Pyrenean rift (ss) by this name.

We agree and changed the sentence to:

In the E, between France and Spain, the Pyrenees sensu stricto (s.s.) formed by continent-continent collision that inverted the Pyrenean extensional system and also propagated into the un-thinned Iberian pro-side (Fig. 1d, Fig. 7c)^{6,7,50}.

310 ‘rifted margin’. Is the Pyrenean depocentre a rift or a rifted margin? So far you have been talking about rifts. If the latter then which margin are your referring to?

We agree and changed to *extensional system*, see sentence above

310 : Suggest propagated would be clearer than ‘progressed’.

We agree and changed accordingly, see sentence above.

313 Add “decreasing from east to west” at end of sentence.

We agree and changed accordingly.

317 : Suggest rewording : “Exhumed fragments of mantle rocks on the N-flank (retrowedge) of the orogen, are found embedded in metamorphosed Mesozoic sedimentary cover between inverted crustal extensional blocks”. I suggest citing not just Teixell et al. (2018) but also, for example, Angrand et al. (2020, J. Geol Soc London) and/or Asti et al. (2022 ; Earth Science Reviews)

We thank the reviewer for a detailed suggestion for rewording and used the proposed sentence. We also added a citation to Angrand et al. 2022 J Geol Soc London. We are already at the limit of possible citations and therefore only chose to cite the field study over the review paper.

320 : In model of Wang et al. the rise of the mantle lithosphere occurred during hyper-extension. This configuration was essentially preserved due to very low convergence in this area of the westernmost north Pyrenees.

Yes, we agree. We are aware of this interpretation, which is not in conflict with the sentence. See also answer below to line 393.

350: inversion?

We prefer to retain the word deformation instead of inversion.

353: no s in highlights

We thank for careful reading, but think that the 3rd person s should be retained:

...the same-polarity orogen.... highlights the complexity

392: Given that the greater Causasus and Pyrenees are both interpreted as mature same polarity orogens, it would be very relevant to discuss the differences between the two orogens in terms of the principal aspects of your models (amount of initial extension? amount of shortening? orientation of linkage zones and branches of rifts?)

We agree that this is interesting. There are many differences between both systems. One of the key differences is that in the Pyrenees lateral differences in orogen structure related to rift inheritance are overprinted by differences in the amount of shortening, as described in the text. We expanded on this sentence as follows:

The Pyrenees are a mature same-polarity orogen, in which lateral differences in orogen structure related to rift inheritance are overprinted by differences in the amount of crustal shortening, which is likely one of the major differences when compared to the Greater Caucasus, another mature same polarity orogen.

We agree that it is very interesting to discuss further differences, also compared to the Atlas, but this discussion is beyond the scope of this study (which is at its length limit), and also beyond the scope of this paragraph, which provides a short summary of the different orogenic systems that we compare to and extends to other orogenic systems.

393 : I dont think this explains the presence of large high density bodies. The offset between the Basque Pyrenees and the main Pyrenees ss can certainly be explained by inversion of a linked offset rift basin. But the presence of high density bodies is surely related to exhumation of deep lithosphere during hyper-extension and their subsequent incorporation into the orogen. In other words their presence is related primarily to the initial style and amount of extension in the rift system...

We thank the reviewer for this comment concerning the two large high density bodies. We agree that other studies have argued that these high-density bodies are related to exhumation of deep lithosphere during extension that was subsequently incorporated into the mountain belt at the inversion stage. However, following this interpretation, we would expect to find these types of large high density bodies along-strike the whole Pyrenees and not only in two specific locations. Yet, we don't. Interestingly, these locations coincide with rift linkage zones. Our models show that inverting an offset linked rift basin can produce these structures in the linkage zones. We think that this is an intriguing result as it provides a very simple alternative explanation. We though also believe that this feature is not the main point of this article and deserves follow-up investigation. To soften our statement, we changed it to:

*Observed large high-density bodies at shallow depth in the Pyrenees can furthermore **potentially** be explained by the inversion of a linked offset rift basin.*

409 : and the amount of shortening?

We agree and added the proposed phrase.

411-414 : Is there any natural mountain belt that demonstrate this behaviour?

Yes, the Hindu-Kush Pamir region is potentially such a mountain belt. The last two sentences of the previous paragraph discuss this, here repeated for clarity:

Subduction polarity reversal has also been proposed for mountain building in the Hindu Kush - Pamir region⁵⁷. The distribution and propagation-direction of shortening⁵⁷, as well as lower topography in the transition zone between both opposite-polarity slabs fits well with the characteristics of an opposite polarity subduction orogen as shown in Fig. 8b.

481 : This section and Figure 9 would be better in my opinion early in the paper to allow the reader to understand the model results. For example, the colours codes for results figures are given on figure 9.

We appreciate the suggestion by the reviewer to move this model setup section into the main body of the article. However, we think that adding this section up front makes the article unnecessarily more technical. Therefore, we would like to retain the section and model setup at the end of the article. However, to make sure that the reader is properly pointed to the Methods section, we added to the first paragraph in section 2.1:

... See Methods section for a detailed model setup description.

We also added model materials legends to the model results figures, which will help for clarity.

Figures

Figure 1. I have located the actual Mauleon basin on figure 1d. As this is a well known and defined local basin, its name is not appropriate for the whole Pyrenean rift system.

We agree and changed the name to *Pyrenean basin*, as proposed by the reviewer in the text.

Figures 2 to 5. It would be helpful to put the compass points on at least the first model image. Need a key or labels for colours on the cross sections.

Can you separate the three cross sections as complex details are not clear in the areas where they overlap ? Same comment for all cross sections.

We thank the reviewer for detailed comments on the figures. We added a compass-arrow to the first landscape plot. We also want to note that the cardinal directions are also displayed on the model setup cartoons in the top right corner of the result figures. We added a legend for the material colours to all plots. We also moved the cross sections a bit. Note that the cross sections and their distance represent the real model dimensions. We want to retain this feature as it is most true to the results. We think that the model figures in addition to the animations show the model development very well now, including details.

Given that the 3D map views are labelled a, b, c, it would probably be better not to use A, B, C for the cross sections ?

We agree and changed to I, II, III

Reviewer #3:

The manuscript titled “Rift linkage and inheritance determine collisional mountain belt evolution“ by Wolf and co-authors presents 3D numerical models of lithosphere extension directly followed by convergence. The main aims of the modelling are to better understand linkage of rift segments during extension and the impact of the rift geometry on the subsequent convergence, particularly the subduction polarity. The applied numerical model consists of a lithosphere deformation model coupled to a landscape evolution model. The study includes a very interesting application of force analysis to explain linkage of rift segments and the subduction polarity during subsequent convergence and rift inversion. Furthermore, the study discusses the application of the model results to the Greater Caucasus, Atlas and Pyrenees. There are still many open questions concerning the impact of 3D features on rifting and rift inversion and the manuscript, hence, presents a very interesting study on the application of deterministic mathematical models to better understand these geodynamic processes. The presented results demonstrate the importance of the pre-orogenic 3D rift geometry on the subsequent orogeny related to the inversion of the rift basins. The authors apply a force analysis to explain the 3D model results which is very useful for interpreting and understanding the numerical results. I find the study of interest for a wide readership in Geosciences and, hence, find the study suitable for publication. I have several comments, with the main aim to make the model results more transparent, which the authors could consider.

We thank the reviewer, Stefan Schmalholz, for a positive and very helpful review. We accommodated the suggestions from the reviewer to make the model results more transparent, as outlined below.

The applied lithosphere deformation model represents an end-member scenario in which the deformation of the crust and lithospheric mantle is dominated by brittle-frictional processes. The maximal friction angles of both crust and mantle lithosphere are only 15 degrees (providing a deeper brittle-ductile transition compared to a friction angle of 30 degree) and only strain weakening due to a reduction of the friction angle with strain is considered which furthermore generates a permanent weakening. No other weakening mechanisms are considered, such as due to grain size variation or shear heating, and no transient weakening mechanisms are considered which would allow for healing and re-strengthening of shear zones and faults that were active during rifting. Since weakening mechanisms other than friction-weakening and healing most likely occur in nature, it is not clear how representative and robust (with respect to model variations by including other weakening mechanisms and healing) the presented results are. Ideally, also another end-member model would be presented with different weakening mechanisms and non-permanent weakening to test how dependent the first order results are on the particular model assumptions. I can imagine that such additional models are outside the scope of the study, but I maybe the authors can at least explain the particularities of their model and discuss the potential impact of different weakening mechanisms and healing on their results.

We thank the reviewer for a very useful comment. We agree that we only use frictional weakening as source of weakness. We only focused on this mechanism, because in field studies, we can observe that brittle-frictional fault zones that formed during basin formation are reactivated during orogenesis, hence likely stay weak even many Myr after extension. Prime-examples are the Pyrenees, inverted Salta rift basin in the Andes, or the Atlas, but really these structures are similarly also observed in all inversion orogens. However, we fully agree with the reviewer that there could be other mechanisms creating permanent or also transient weakening, like grain-size variations, shear heating, or metamorphic phase changes creating a dominant metamorphic fabric. Indeed, the directional weakness needed to create same-polarity subduction in the Pyrenees might be related to a pre-existing Hercynian metamorphic fabric, as described in the text. We believe that running simple models and properly quantifying them, as done in the study, is a good approach and provides transferability of results. We believe that running additional models with different types of weakening is beyond the scope of this study. A proper exploration of all different types of weakening, including (partial) healing would be work for several additional articles. The models presented in this study can serve as a comparative baseline for such additional studies. To account for this discussion, we added the following paragraph to the model limitations section:

We only apply frictional-plastic softening as a weakening mechanism in our models. Although the reactivation of weak frictional-plastic shear zones is characteristic of inversion orogens - such as those examined in this study - we note that additional permanent or transient weakening processes may play a role during extension and orogenesis, like grain-size variations, shear heating, or metamorphic phase changes. Future modelling efforts should explore these additional mechanisms. The model results and force balance analysis presented here can serve as a comparative baseline for such investigations.

For the force analysis, it would be useful to also know the values of the depth-integrated strengths, for example F_{weak} and F_{strong} (in N/m ; discussed in the supplementary information) and how they vary laterally because these values could then be compared to other estimates for these strengths, such as plate driving forces (e.g. slab pull, ridge push or force balance at continental plateaus). It would be also useful to have figures that show the cross section of the models with the corresponding deviatoric stress field to be able to see the stress magnitudes and ideally the brittle-to-ductile transition in the models. Similarly, it would be useful to have additional figures of the cross sections that indicate with a colormap which regions deform in the brittle-plastic regime and which ones in the viscous regime. This can further help to compare the model results with field observations or earthquake observations. Such figures could be put in the supplementary material. Currently, there are no figures that show any stress fields, although they are the fundamental values used in calculating the forces, and it is not clear how the brittle-frictional and viscous deformations are distributed during orogeny.

We thank the reviewer for this very useful comment. We now report the required values. Here a few notes about these forces. We considered computing F_{weak} and F_{strong} and particularly their integration along strike of the modelled orogen directly from the deviatoric stress fields of the models. However, we could not find a good criterion to do this computation automatically, for instance based on a strain rate-cutoff. Rather, we found that computing F_{weak} and F_{strong} analytically and integrating along simple

distances as shown in the block diagrams (Fig. 6) gives a very good approximation of the model behaviour. For the computations we used a constant strain rate of $1e-14$ 1/s that is typical for the deformation rates in our models, assumed lithostatic pressure, and assumed a steady-state thermal field corresponding to the initial setup of the model. Using these simplifications, we found that the transition from same-polarity subduction (seed offset 0 – 30 km) to opposite polarity subduction (seed offset > 30km) can be predicted very accurately by the analytical solution when compared to the models. Similarly, the predicted transition from basin-linkage to no linkage during extension (at 292 km) also fits very well with what we observe in the models. This tells us that despite these simplifications, the analysis works well. Assuming lithostatic pressure is a simplification, as it neglects tectonic overpressure stemming from gravitational potential energy variations. We tested the effect of overpressure through simple multiplication of the lithostatic pressure. We found that it had no first-order influence on the analytical results, but would of course increase F_{weak} and F_{strong} . For instance, a factor 2 on the lithostatic pressure only changed the basin linkage transition from 292 km to 290 km, which is secondary. We believe that presenting a simple analytical solution that is easy to understand and successfully predicts model behaviour is useful and better than a very complex computation that has no gain in predictive power. We added the following text to the supplementary material:

Assuming lithostatic pressure conditions, a constant typical strain rate of $1e-14$ s⁻¹, and a steady-state thermal field corresponding to the initial thermal field of the models gives $F_{weak} = 2.8e12$ N/m and $F_{strong} = 12.1e12$ N/m. These assumptions neglect tectonic overpressure, e.g. related to gravitational potential energy variability, which can be significant [Schmalholz et al., 2014]. Sensitivity tests including substantial overpressure showed no first-order effects on the resulting relationships for basin linkage or subduction polarity, but would increase F_{weak} and F_{strong} . These tests suggests that the analytical results are robust within the framework of our assumptions, and show that the reported values of F_{weak} and F_{strong} are minimum values.

The citation to [Schmalholz et al., 2014] was also added to the main manuscript.

We agree with the reviewer concerning the additional plots and added model plots of the deviatoric stress fields as well as plots of the deformational regimes. These plots are shown in the supplementary material as figures S1-S4. We think that these are nice and useful plots. We note that the fields were computed during post-processing, with one average value per cell. In pTatin3D, properties are computed at each Lagrangian marker particle and then averaged over a cell. Because of this averaging, we expect very small differences between the fields presented here and the fields used for computation in pTatin3D. The latter three sentences are also stated in the supplementary material. We also added the following sentence to the Results section:

Supplementary model output showing deformational regimes and deviatoric stresses is presented in Supplementary Figures S1 - S4.

Some minor comments and questions:

Line 75 : Please provide the formula for the B_m number. Also, please provide the values used in the simulations; $B_m > 1$ is vague.

We agree and provide the value of $B_m = 3.5$ in line 75. Its computation involves a number of variables so that we moved the computation to the Methods section and added a sentence to line 75 referring to the methods section. See also answer below.

Does the landscape evolution model has a significant impact on the collisional model evolution or would a simpler erosion model yield similar results?

This is an interesting, yet rather hypothetical discussion, because we did not try with other landscape evolution models. $B_m = 3.5$ implies that the model is “tectonics dominated”, i.e. the orogen is widening and in height limited by the lithospheric strength and not the efficiency of erosion, which seems to be the norm on Earth. Therefore, we speculate that the observed subduction pattern and first order evolution of mountain building is likely also happening in models without surface processes or with a simpler erosion model, e.g. approximated by diffusion, as long as tectonics is “dominant” over erosion. However, the sedimentation pattern and diagnostic landscape evolution pattern with river valleys being a marker of deformation requires a landscape evolution model similar to the one used here. In that sense, having a landscape evolution that “properly” parametrizes erosion and sediment transport is important and very useful. Although interesting, we don’t think this discussion needs to be in the manuscript, as it is tangential to the work and rather hypothetical.

2.2.1. : As mentioned, it would be useful to know also the values of the vertically integrated strengths to compare them with plate driving forces (in N/m).

Yes, we agree and report these in the supplementary material, see also answer above. We added the following to this paragraph to highlight the values:

*See supplementary information for detailed information on the definition of the different forces involved, **including characteristic values.***

2.2.2. : The model assumes significant permanent weakening (minimum friction angle of 2 degree in both crust and mantle) and no healing. Hence, it is expected that in the presented models the permanent weak zones developed during rifting control the subsequent collision stage.

We agree with the reviewer. We added a discussion of healing in the model limitations section, see answer above. Specifically, we also added the following sentence to the first paragraph:

Further studies should nonetheless investigate the effect of post-rift cooling in detail, including an investigation of transient weakening effects including healing mechanisms.

Equation (3) : It seems that shear heating due to dissipative deformation (friction and viscous shearing) is not considered? If associated shear heating by dissipative deformation is not considered in equation (3), then energy is not balanced in the model.

Yes, we agree. This shortcoming is discussed in the model limitations section. We did test the importance of shear heating in very similar 2D models, and could not find a first order effect on model results. We therefore expect no first order changes to the model

results when including shear heating in the models presented here, but would do so in the future.

Equation (4) : The model is 600 km deep. Why density is not also a function of pressure?

We opted to not include density as function of pressure, because the deformational domain of interest is the upper 200 km, where changes as function of temperature and composition are much more important than changes as function of pressure, which are secondary. We agree though that this could be added. We added the following sentence:

We do not include changes in density as function of pressure.

Line 439 : Strain weakening by reducing the friction angle is usually mesh-dependent if no regularization method is applied. How is the mesh-dependence handled in the numerical model?

We agree with the reviewer that this is a shortcoming of the simplified strain-weakening used. We note that this parametrisation is nonetheless commonly used throughout the community and reported in the literature. We tested different strain weakening thresholds and found that they did not have a first-order impact on model results. We added the following sentence:

We do not apply regularization methods to eliminate mesh-dependence of the strain-weakening algorithm. Tests with different strain weakening thresholds did not show a first-order effect on model results.

Line 466 : Please provide the formula for the Bm number.

We agree, and added the formula of the Bm number as well as an explanation of its variables.

Line 505 to 509. When the lithosphere is subducted into the sub-lithospheric upper mantle, how can this parameterization of k then produce a correct temperature field around the subducting lithosphere? This parameterization may work well during the extension phase, but how can it be justified for the subduction process?

We thank the reviewer for this comment. As written in the text, we use this parametrisation to *maintain the heat flow at the base of the lithosphere, prevent the model from cooling, and resembling mantle convection at high Nusselt number*. We believe that this a very useful parametrisation, as it mimics a natural process and keeps constant thermal conditions in the lithosphere. We agree that when modelling whole-mantle subduction, then this parametrisation is not needed. However, our models do not exhibit deep lithosphere subduction. Rather, the lithosphere subducts at most to depths of 200 – 250 km, and only reaches this depth during the late stages of model evolution. Therefore, we opted to use this parametrisation for this study.

Line 534 to 537 : This limited effect is controlled by the model assumptions because only a frictional strain weakening mechanism is considered which causes permanent weakening. No transient weakening mechanisms are considered which would allow healing and re-

strengthening rift-related faults and shear zones. Therefore, the limited impact of a cooling phase is model related, but not related to natural processes which can involve healing mechanisms.

We agree and changed the last sentence of this paragraph to:

Further studies should nonetheless investigate the effect of post-rift cooling in detail, including an investigation of transient weakening effects including healing mechanisms.

We note though that in orogenic systems that had significant amounts of post-extension quiescence, like the Atlas, we still observe that extensional fault zones are preferentially reactivated during shortening. This indicates that they do not completely heal. Exploring this in detail is though beyond the content and scope of this article.

Best regards,
Stefan Schmalholz

References:

Schmalholz, S. M., S. Medvedev, S. M. Lechmann, and Y. Podladchikov (2014), Relationship between tectonic overpressure, deviatoric stress, driving force, isostasy and gravitational potential energy, *Geophys J Int*, 197(2), 680-696, doi:10.1093/gji/ggu040.

Response to Reviewers for final revision

Sebastian G. Wolf, Ritske S. Huismans, Josep Anton Muñoz, Dave A. May

This document concerns our final revision of the paper entitled *Rift linkage and inheritance determine collisional mountain belt evolution*. We thank all three reviewers again for very positive, constructive, and useful comments that helped improve our manuscript and are happy that the reviewers considered our revisions suitable. We accommodated the few remaining changes suggested by the reviewers. Below follows a point-by-point reply to the reviewer comments. Comments are in black font with serif; our answer is written in blue font without serif. Lighter blue text without serif highlights changes to the manuscript text.

Reviewer #1

The authors have adequately addressed all of my comments and concerns in their revised manuscript. I find the current version suitable for publication and recommend acceptance in its present form.

We thank the reviewer for useful comments and are pleased to hear that the reviewer considers her/his comments and concerns adequately addressed.

Reviewer #2

I have read through the revised manuscript and the reply of authors to reviewers. The authors have carefully addressed and discussed the comments and revisions suggested by reviewers. The paper is now much improved in clarity and represents an important and illuminating contribution to our understanding of rift inversion and orogenesis at a lithospheric scale.

One minor correction needs to be made on figure 1d - change Mauleon Basin to Pyrenean Basin as revised in the text.

I congratulate the authors on this very nice work.

I recommend publication

Mary Ford, Université de Lorraine, Nancy, France.

We thank the reviewer, Mary Ford, for detailed reading and useful comments, and are pleased to read that our revisions are considered suitable. We agree with the reviewer and changed the basin name on figure 1d to Pyrenean Basin as suggested.

Reviewer #3:

The authors have addressed my comments and clarified the text with respect to model assumptions. They also added new panels in the figures in the supplementary material showing the deviatoric stress which I find very useful. Only in case the authors will make

another revision (which is not necessary), I have few minor comments to the manuscript. I also have two comments to their rebuttal.

We thank the reviewer, Stefan Schmalholz for useful constructive feedback and are pleased to read that he considers further revisions not necessary. We nonetheless accommodated the minor comments by the reviewer, see below.

Minor comments

Line 377: "...if their offset is not too large..."; maybe the authors could quantify what is "not too large" with a dimensionless number, for example, if the ratio of offset to crustal thickness is less than approximately...

We thank the reviewer for this comment. We quantified "what is not too large" in section 2.2 under *Continental rifting: To link or not to link*. There we show that linkage depends on distance between the weak zones and their relative lithospheric strength. It's thus not only one factor that can be easily described by a non-dimensional number. We agree though that this sentence can be clarified and changed it to the following sentence:

(i) extensional basins with closest proximity and lowest lithospheric strength are most likely to link during rifting

Line 451: If H is only the radiogenic heat production rate, please specify this.

Yes, it is. We followed the reviewer suggestion and added the word **radiogenic** to the respective sentence.

Line 510: For completeness, the authors could also specify the value or criteria for the mechanical time step.

We agree and added the following sub-sentence:

..., which is dynamically computed based on a Courant–Friedrichs–Lewy condition and typically in the order of 9000 yr.

Comments to rebuttal and new text in supplementary material

Lithostatic pressure and force balance analysis: It is still not clear to me why the authors assume lithostatic pressure and a constant strain rate for the force calculation. Why don't the authors not use the values of deviatoric stresses calculated in their 3D thermo-mechanical model to calculate F_{int} ?

We thank the reviewer for this comment, which we split in two for simplicity. As written in the previous reply, we tried to compute F_{weak} and F_{strong} directly from the model. This is straight forward for F_{strong} , as it can be computed with a 1D depth-integrated strength profile. It is, however, not a simple 1D profile for F_{weak} , which would need to follow a 3D path along a shear zone. Doing this numerically is possible, but comes with additional complexity, e.g. which criteria to use in choosing the path along which to integrate. Therefore, we opted to use yield-strength envelopes. We then tested whether

this simple approach can reproduce the observed deviatoric stresses when we simply scale the lithostatic pressure by a factor, and find that our approach is a useful first-order approximation. Furthermore, as written in the previous reply, the resulting values that we get for instance for the maximum offset for linkage of offset rift basins (292 km) are not sensitive to the overpressure-factor used. Our calculations are also successful in predicting model behaviour so that we consider them useful. We think that this point is already discussed in the supplementary material. To point the reader more clearly to the supplementary material, we amended the following sentence to the manuscript (addition in bold font):

*See supplementary information for detailed information on the definition of the different forces involved, including characteristic values **and simplifying assumptions**.*

Concerning deviations from lithostatic pressure: figures S1 to S4 all indicate deviatoric stresses of several hundreds of MPa up to 1 GPa according to the colorbar. Since tectonic overpressure magnitudes are similar to deviatoric stress magnitudes, overpressure should be significant in the models. Therefore, if there are significant deviatoric stresses, then the assumption of lithostatic pressure is not justified. In the rebuttal, the authors say that tectonic overpressure stems from gravitational potential energy (GPE) variations. However, this applies only to situations without far-field deformation. In a lithosphere under compression/shortening, there can be significant tectonic overpressure also without GPE variations if deviatoric stresses are large, as is the case in the presented models. For the yield strength envelope for F_{strong} , shown in figure 6, there is significant overpressure, for example, in the brittle regions of the mantle lithosphere. This has been shown, for example, in the study of Petrini & Podladchikov, Journal of metamorphic Geology, 2000.

We thank the reviewer and fully agree with his assessment that there are significant overpressures. As described above, we assume lithostatic pressure and use a scaling factor to approximate overpressure conditions. We think that this is a simple and useful approach, and discussed in the supplementary material. We agree that overpressure also arises from lithosphere being under compression/shortening as for instance described by Petrini & Podladchikov, 2000. Therefore, we wrote in the supplementary material:

These assumptions neglect tectonic overpressure, e.g. related to gravitational potential energy variability, which can be significant [Schmalholz et al., 2014]

The “e.g.” should indicate that there are other “sources” of overpressure, without going into a longer discussion. We nevertheless think that it is useful to mention far-field deformation effects and that overpressure is observed in our models, and changed the sentence above (additions in bold font):

*These assumptions neglect tectonic overpressure, e.g. related to gravitational potential energy variability **or far-field deformation**, which can be significant [Petrini and Podladchikov, 2000; Schmalholz et al., 2014] **and is also observed in our models (Figs. S1 - S4)**.*

Response concerning line 439: “We note that this parametrisation is nonetheless commonly used throughout the community and reported in the literature.“. This is not a convincing reply, because the fact that others use a certain method does not necessarily mean that the

method is correct. I am also not sure which community is meant. The mesh-dependence can affect the calculated load-bearing capacity of a strain-softening material and hence can affect the force calculation. But since the authors do anyway not use their 3D model results, involving strain softening, to calculate the forces, but a simplified 1D yield strength envelope, their force analysis is not affected by the mesh-dependence.

We thank the reviewer for this comment. As the reviewer writes, it is not affecting the force analysis. With the community we mean other research groups that also investigate upper mantle-scale deformation with non-linear rheologies and large strain. Many of them, using also other codes than the one used here, use strain-dependent weakening of the frictional parameters, which is the same approach.

The overall results of the force balance analysis make sense, but the accuracy and numerical robustness of the calculated values of F_{weak} and F_{strong} (in units of N/m) and the resulting forces (in units N) is not clear to me.

We thank the reviewer for this comment. As discussed in the supplementary material, the presented values are minimum estimates because of the assumption of lithostatic pressure. We think that the additional supplementary model plots give a good overview of the involved deviatoric stresses.

Best regards,
Stefan Schmalholz

References:

Petrini, K., and Y. Podladchikov (2000), Lithospheric pressure-depth relationship in compressive regions of thickened crust, *J Metamorph Geol*, 18(1), 67-77.

Schmalholz, S. M., S. Medvedev, S. M. Lechmann, and Y. Podladchikov (2014), Relationship between tectonic overpressure, deviatoric stress, driving force, isostasy and gravitational potential energy, *Geophys J Int*, 197(2), 680-696, doi:10.1093/gji/ggu040.